# A shared mechanism for Bacteroidota protein transport and gliding motility

Xiaolong Liu ®[1] ✉, Marieta Avramova ®[1], Justin C. Deme ®[2], Rachel L. Jones ®[1], Camilla A. K. Lundgren ®[1], Susan M. Lea ®[2,3] ✉ & Ben C. Berks ®[1] ✉

Bacteria of the phylum Bacteroidota are major human commensals and pathogens in addition to being abundant members of the wider biosphere. Bacteroidota move by gliding and they export proteins using the Type 9 Secretion System (T9SS). Here we discover that gliding motility and the T9SS share an unprecedented mechanism of energisation in which outer membrane proteins are covalently attached by disulfide bonds to a moving internal track structure that propels them laterally through the membrane. We determined the structure of an exemplar Bacteroidota mobile track by obtaining the cryoEM structure of a 3 MDa circular mini-track from *Porphyromonas gingivalis*. Our discoveries identify a mechanistic and evolutionary link between gliding motility and T9SS-dependent protein transport.

The Bacteroidota (formerly Bacteroidetes) are one of the three main phyla of Gram-negative bacteria. They are normally the most abundant Gram-negative commensals in the human gut and are also major components of other human microbiota[1]. The phylum includes major human opportunistic anaerobic pathogens responsible for sepsis, as well as oral pathogens such as *Porphyromonas gingivalis* which cause severe dental disease (periodontitis). More broadly, the Bacteroidota are widely distributed in environments throughout the biosphere[2]. Two key distinctive features of the Bacteroidota are the ability to move using gliding motility and to export proteins using the Type 9 Secretion System (T9SS).

The T9SS transports substrate proteins across the outer membrane (OM) and plays critical roles in Bacteroidota virulence and nutrition[3–9]. During Type 9 transport, the substrate protein moves through a channel in the OM translocon protein SprA to form a complex with a docked OM carrier protein[10] (Fig. 1a). The resulting substrate-carrier protein complex is then ejected from the translocon using an energetic input from a multi-protein Energy Chain[11]. T9SS substrates fall into two classes[12]. Type A substrates are extracted from the translocon by the general carrier protein PorV and are subsequently either released into the extracellular environment or covalently attached to cell surface lipopolysaccharide[11,13]. By contrast, Type B substrates each have their own dedicated carrier protein to which they remain bound thereby anchoring them to the cell surface[14,15].

Gliding is the only mode of motility present in the Bacteroidota and allows rapid movement across solid surfaces[7,16]. Genomic analysis suggests that gliding motility is present in most free-living Bacteroidota and some oral commensals[17,18]. Bacteroidota gliding is mechanistically distinct from the gliding motility exhibited by other phyla[16]. Bacteroidota gliding motility has been best studied in the soil bacterium *Flavobacterium johnsoniae*. Gliding in this organism is mediated by the cell surface adhesin SprB, which flows in helical trajectories up and down the cell[19–23]. When SprB adhesins attach to a solid surface, the cell is propelled forward in a cork-screwing motion[10,21,22,24]. The SprB adhesin is anchored to the cell surface by the OM protein SprF[14,15,25] (Fig. 1a). SprF is, in turn, thought to bind to a moving track structure at the inner face of the OM, resulting in the propulsion of the attached adhesin along the cell body[18,22,26,27]. In current models of the gliding apparatus the track is predominantly, or exclusively, a polymer of the OM-anchored lipoprotein GldJ[22,26,28].

Bacteroidota gliding has long been known to depend on the organism possessing a T9SS in order to transport the gliding adhesins to the cell surface[29]. For example, in *F. johnsoniae* the adhesin SprB is a Type B T9SS substrate and SprF is its cognate carrier protein[14,15]. More recently, it has been shown that gliding motility and Type 9 transport are intimately mechanistically linked because they are both powered by the T9SS Energy Chain[23,29,30]. In current models the Energy Chain is composed of motor complexes that transduce the energy of the inner

[1]Department of Biochemistry, University of Oxford, Oxford, UK. [2]Center for Structural Biology, Center for Cancer Research, National Cancer Institute, Frederick, MD, USA. [3]Structural Biology, St Jude Children's Research Hospital, Memphis, TN, USA. ✉e-mail: xiaolong.liu@bioch.ox.ac.uk; susan.lea@stjude.org; ben.berks@bioch.ox.ac.uk

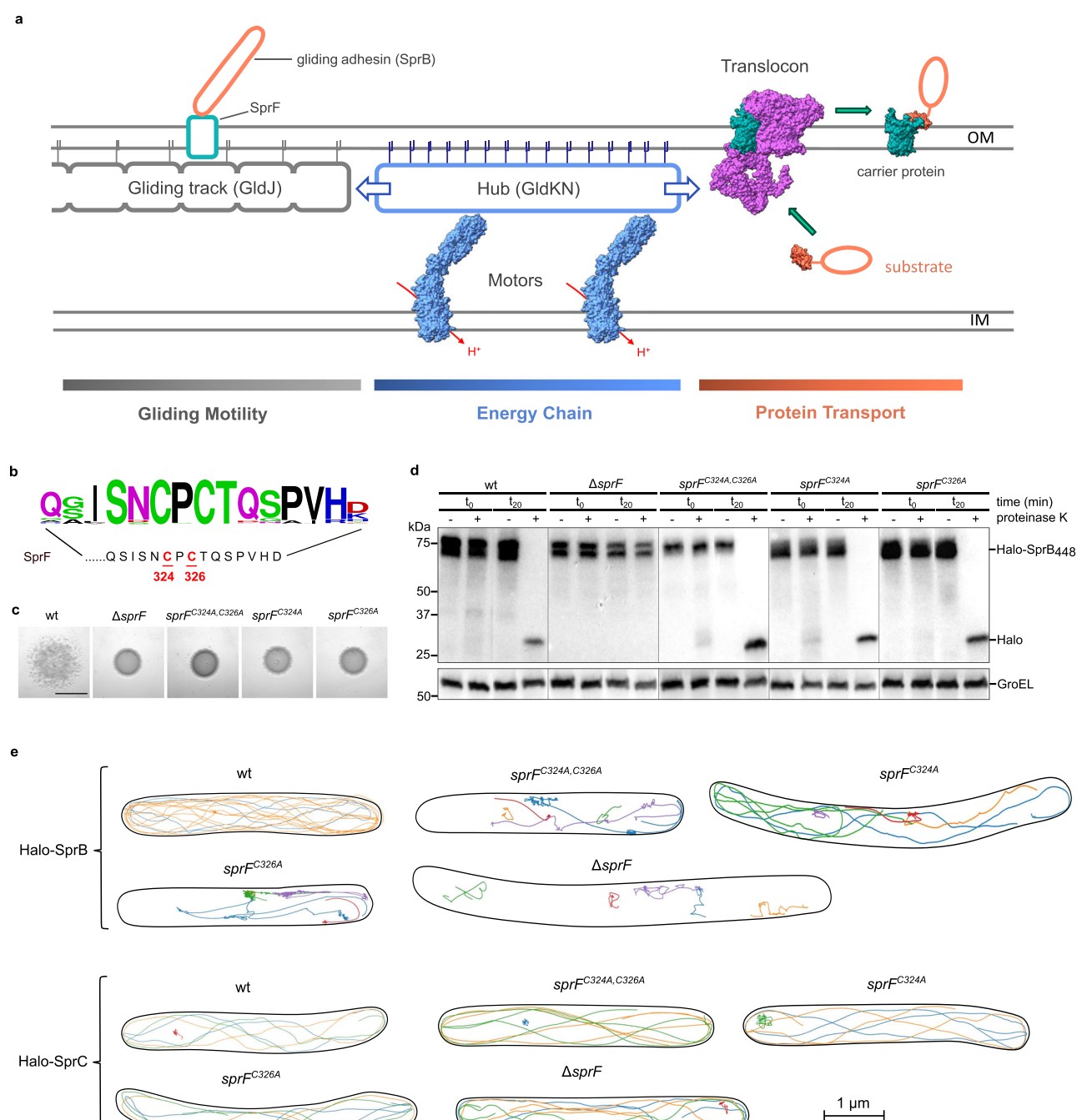

**Fig. 1 | The cysteine residues in the periplasmic C-tail of SprF are important for SprB movement and gliding motility. a** Current model for the organization of the gliding motility apparatus and T9SS in *F. johnsoniae* involving a shared Energy Chain. OM, outer membrane. IM, inner membrane. **b** Sequence of the periplasmic C-tail of *F. johnsoniae* SprF with the twin-cysteine motif highlighted and a Weblogo display of the sequence conservation (500 closest whole protein homologs) above. **c**–**e** Characterization of *sprF* mutants with substitutions in the conserved cysteine pair. wt, wild type. Similar results were obtained for 3 biological repeats. **c** Spreading (gliding) morphology on agar. Scale bar, 5 mm. **d** Cell surface exposure of the adhesin SprB assessed by protease protection. The indicated strains expressing a fusion between HaloTag and the final 448 amino acids of SprB (Halo-SprB$_{448}$) were incubated with Proteinase K ( + lanes). Reactions were stopped immediately ($t_0$) or after 20 min ($t_{20}$) and analysed by immunoblotting with Halotag antibodies. The cytoplasmic protein GroEL serves as both a loading and cell integrity control. In each panel (which correspond to separate immunoblots) a lane has been removed between the $t_0$ and $t_{20}$ samples. **e** Exemplar trajectories of single fluorophore-labeled HaloTag-SprB adhesin molecules or HaloTag-SprC molecules in the indicated backgrounds. The trajectories link the single molecule localizations in successive frames recorded for agarose-immobilized cells and are 2D projections of what is a 3D process. Each trajectory within a cell is shown in a different color and the segmented outline of the cell is shown in black. For the SprF cysteine variants cells were chosen to illustrate the observed range of SprB mobility behaviors. See also Supplementary Movie 1.

membrane protonmotive force into mechanical energy and transfer this across the periplasm to a Hub complex on the inner face of the OM[23,31,32] (Fig. 1a). The Hub complex then distributes this mechanical energy both to the T9SS translocon and to the gliding machinery to power movement of the gliding track[23]. In the non-gliding pathogenic bacterium *P. gingivalis* the Hub has been shown to be a megadalton ring-shaped molecule composed of many copies of the proteins PorK and PorN, and anchored to the OM by the lipidated N-terminus of PorK[29,33–35]. In the gliding bacterium *F. johnsoniae* the PorKN-equivalent Hub proteins GldK and GldN[36] are assumed to form a similar ring

structure, the outer rim of which contacts the gliding track[16,28]. The motors are envisaged to rotate the Hub ring (pinion) to drive the movement of the associated gliding track (rack)[27,28].

If the adhesin carrier protein SprF directly contacts the gliding track, as proposed, then it would be anticipated that this interaction would need to be extremely strong given the enormous mechanical forces exerted on the adhesins during gliding. However, we were unable to predict a plausible complex between SprF and the presumed track protein GldJ using AlphaFold3 (AF3)[37]. Intrigued by this observation, we decided to investigate the nature of the connection between SprF and the track.

Here, we uncover an unanticipated and shared role for disulfide bonds in linking gliding and T9SS components to moving internal track structures. This work leads to a radically changed mechanistic model for the gliding motility apparatus and its relationship to the T9SS.

## Results

### Twin cysteines in the C-tail of SprF are important for gliding motility

In considering how the adhesin carrier protein SprF might connect to the gliding track, we noticed that the periplasmic C-terminal tail of SprF contains a Cys-Xaa-Cys motif that is invariant in all close homologs (Fig. 1b). Since conserved cysteine residues often have a functional role, we examined whether this twin-cysteine motif is required for SprF to support gliding motility. Substituting either or both of the cysteine residues abolished gliding on agar (Fig. 1c). Thus, both cysteine residues are important for SprF function. Control experiments confirmed that the cysteine substitutions do not affect the expression level or stability of SprF (Supplementary Fig. 1).

SprF can be considered to carry out two mechanistically separable functions. First, it acts as a carrier protein for the export and surface-anchoring of the SprB adhesin. Second, it attaches to the periplasmic gliding tracks to allow movement of the bound adhesin. We used protease accessibility experiments to test whether the twin-cysteine motif is required for SprF to fulfill its carrier protein function (Fig. 1d). In these experiments, a protease is added to the medium around the cells to digest any SprB molecules present on the cell surface. However, if export of SprB is unable to take place (for example, in a ΔsprF background in Fig. 1d), the adhesin remains inside the cell and is protected from digestion by the external protease. Using this approach, we found that substitution of the SprF cysteine residues either singly or in combination had no effect on the surface localization of SprB (Fig. 1d). Thus, the twin-cysteine motif is not required for the carrier protein function of SprF.

To determine whether the twin-cysteine motif is, instead, required for the interaction of SprF with the gliding track we carried out live cell single molecule tracking experiments (Fig. 1e, Supplementary Movie 1). In this approach fluorophore-labeled SprB adhesin molecules are tracked as a proxy for the behavior of the SprF molecules to which they are bound. In a wild-type SprF background a large proportion of the SprB molecules move on sustained helical trajectories as previously reported[19–23]. However, substituting the cysteine pair in SprF almost completely suppresses this continuous helical movement. Thus, the twin-cysteine motif is required for SprF to make normal interactions with the gliding track. Although most of the SprB molecules in the cysteine-substituted background were close to stationary, a minority still exhibited short, apparently directed motions. 3D tracking confirmed that the handedness of these short trajectories matched the exclusively left handed geometry of the gliding tracks[20,22] showing that these trajectories still represent SprB molecules moving on a track (Supplementary Fig. 2). The adhesins with these short trajectories showed frequent stopping behavior, often for extended periods of time (> 10 seconds). During these stops the molecules could be either immobile or slowly diffusive, and the adhesin often resumed helical

movement in a different direction from the trajectory before the stop. Thus, SprF is still able to interact with the gliding tracks in the absence of the twin-cysteine motif, but these residual interactions are sporadic, short term, and prone to aberrant behavior. Control experiments show that if SprB export is blocked by the complete removal of SprF (ΔsprF) then the non-exported SprB molecules in the periplasm exhibit rapid isotropic diffusion that is quite distinct from the behaviors observed when the SprF cysteine residues are removed.

### SprF is disulfide-bonded to GldK

Conserved cysteines residues in periplasmic proteins are often involved in forming disulfide bonds. This prompted us to explore whether the twin-cysteine motif in SprF was involved in forming a disulfide link to the gliding track. To do this, we first trapped the thiol/disulfide state of the cellular proteome by treating F. johnsoniae with the membrane-permeable thiol-reactive reagent N-ethylmaleimide (NEM). NEM forms an irreversible adduct with free thiol groups. The modified thiols are no longer able to attack and break disulfide bonds and so the disulfide bonds present in the cell are preserved.

Treatment of F. johnsoniae cells with NEM resulted in almost the entire SprF pool being recovered in a high molecular mass adduct running larger than 100 kDa (Fig. 2a). Formation of this adduct was reversed by the addition of the thiol-containing reagent DTT, confirming that the two proteins were linked by a disulfide bond (Fig. 2a). Formation of the adduct also required the presence of the twin-cysteine motif in SprF, demonstrating the involvement of these residues in the disulfide linkage (Fig. 2a).

SprF anchors the gliding adhesin SprB to the cell surface. We found that SprB was required for the formation of the SprF-containing disulfide adduct (Fig. 2a). The disulfide adduct was also missing in a strain in which export of SprB was blocked by deletion of the T9SS translocon (ΔsprA strain, Fig. 2b), though not by removal of a T9SS carrier protein that is not involved in SprB export (ΔporV strain, Fig. 2b), confirming that SprB must be exported and loaded onto SprF for disulfide bond formation to occur. The stability of the SprF adduct does not depend on active movement of the SprB-SprF complex since it is still detected after dissipation of the protonmotive force that powers the gliding apparatus[20,23] (Supplementary Fig. 3).

To identify the partner protein to which SprF is disulfide linked, we took a candidate approach informed by our current understanding of the organization of the gliding motility apparatus. In this model, the periplasmic domain of SprF interacts with a gliding track constructed from GldJ protomers (Fig. 1a). Based on this model, the partner protein to which SprF is disulfide-linked should be GldJ. Surprisingly, however, no adduct equivalent to that identified with SprF antibodies was detected by immunoblotting against GldJ (Fig. 2a). This rules out GldJ as the binding partner of SprF. Upon broadening our screen to other gliding components, we found that the lipoprotein GldK was partially recovered in a disulfide-linked adduct with the same molecular mass as that formed by SprF (Fig. 2a). Crucially, this adduct was no longer formed when the twin-cysteine motif of SprF was removed, showing that SprF is the interacting partner (Fig. 2a). Thus, GldK is the partner protein to which SprF is disulfide linked. This conclusion was verified by co-purification experiments under denaturing conditions, which prove that GldK is covalently linked to SprF through a disulfide linkage (Fig. 2c, Supplementary Fig. 4). Since SprF appears to be in continuous movement along the gliding track, the covalent link between SprF and GldK implies that GldK is a component of the track rather than being located in a separate Hub complex as envisaged in the current gliding model (Fig. 1a).

We next attempted to identify the cysteine residue(s) in GldK that are involved in forming the disulfide linkage with SprF. Discounting the lipidated N-terminal cysteine, GldK contains two cysteine residues (Cys285 and Cys343). These cysteine residues are highly conserved across GldK homologs (Fig. 3a). Gliding motility is blocked when

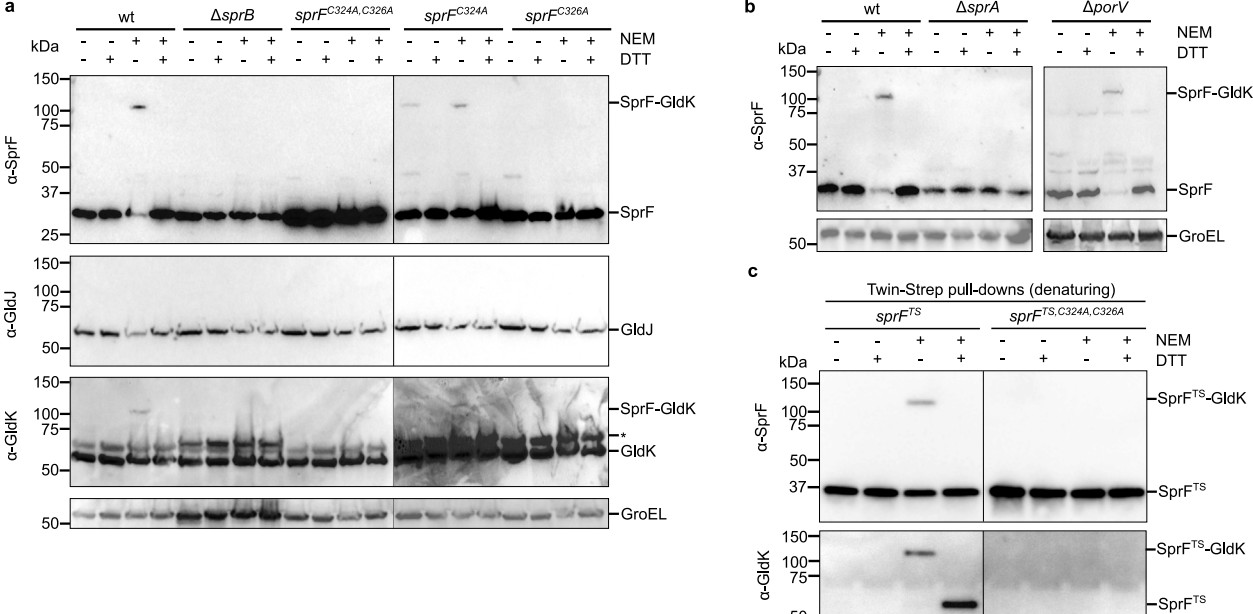

**Fig. 2 | SprF forms a disulfide bond with GldK. a, b** Strains with the indicated mutations were subject to whole-cell immunoblotting with the specified antisera. Where indicated, the cells were pre-treated with NEM. At the SDS-PAGE step, samples were run under either non-reducing (- DTT) or reducing (+ DTT) conditions. GroEL serves as a loading control. *, non-specific background band. wt, wild type; a strain missing the T9SS translocon (ΔsprA); a strain missing the Type A carrier protein PorV (ΔporV). Similar results were obtained for three biological repeats. **a** SprF forms disulfide bonds with GldK but not GldJ. **b** Formation of a disulfide bond between SprF and GldK depends on Type 9 transport activity but is independent of Type A T9SS export. **c** GldK co-purifies with SprF through a disulfide linkage. Twin-Strep-tagged SprF (SprF^TS) variants from whole cells that had been pre-treated or not with NEM as indicated were isolated under denaturing conditions using Streptactin-coated beads. The bead-bound proteins were subject to immunoblotting with the specified antisera under either non-reducing (- DTT) or reducing (+ DTT) conditions. Similar results were obtained from 2 biological repeats.

Cys343, but not Cys285, is substituted providing presumptive evidence that it is Cys343 that forms the disulfide linkage with SprF (Fig. 3b). This is confirmed by biochemical analysis which shows that the SprF-GldK disulfide bond is still present in a *gldK^C285S* background but missing in a *gldK^C343S* strain (Fig. 3c). Export of the SprB adhesin still takes place when the disulfide bond-forming Cys343 of GldK is substituted, albeit with lower efficiency than in the wild-type strain (Fig. 3d). Nevertheless, as with the SprF twin-cysteine deletion strain, single particle tracking measurements fail to detect SprB molecules following the sustained helical trajectories typical of gliding adhesins in wild-type cells. Although there are rare examples of adhesins exhibiting short helical trajectories, most adhesins are close to stationary (there is also a background of rapidly isotropically moving particles due to the elevated levels of non-exported SprB molecules in the periplasm of this strain)(Fig. 3e, Supplementary Movie 2). Thus, for those adhesins that have been successfully exported, the effects of substituting GldK Cys343 on their mobility are equivalent to those of substituting the SprF twin-cysteine motif. This is as expected if these two groups of cysteine residues are partners in the same disulfide bond.

**The SprF twin-cysteine residues have non-identical roles**

We next investigated the individual contributions of the two SprF Cys residues to disulfide bond formation with GldK. Substituting Cys326, but not Cys324, prevented formation of the adduct with GldK (Fig. 2a). This shows that it is Cys326 that forms the disulfide bond to GldK (Fig. 3f). Although the SprF-GldK disulfide bond still forms when Cys324 is removed, the bond is only formed by a minority of the SprF molecules present. This contrasts with the situation with the wild-type protein, where almost the entire SprF pool is crosslinked to GldK (Fig. 2a). Thus, Cys324 enhances the efficiency of formation of the inter-protein disulfide bond. Our interpretation of this observation is that Cys324 forms an intramolecular disulfide bond with Cys326, and

that this bond is then attacked by a cysteine thiol on GldK to form the SprF-GldK disulfide link (Fig. 3f). Consequently, when the intramolecular disulfide bond in SprF is absent in the SprF Cys324Ala variant, formation of the disulfide bond between the remaining SprF cysteine and Cys343 in GldK requires an adventitious oxidant and so occurs with low efficiency.

It is notable that the SprF-GldK adduct is present even without NEM treatment of the bacteria in cells expressing the SprF Cys324A variant (Fig. 2a). This suggests that Cys324 is responsible for the instability of the SprF-GldK disulfide linkage in wild-type cells, which it presumably attacks to reform the intramolecular Cys324-Cys326 disulfide bond on SprF (Fig. 3f). Because NEM treatment of wild-type cells stabilizes almost the entire SprF pool in disulfide linked adducts with GldK (Fig. 2a), we can infer that under physiological steady state conditions Cys324 is present almost exclusively in the thiol state (and thus derivatized by NEM) and so the Cys324-Cys326/Cys326-GldK disulfide equilibrium is positioned heavily towards the Cys326-GldK side (Fig. 3f). This is intuitively reasonable because the vicinal thiols in SprF are separated by only a single amino acid residue and so the disulfide bond between them will be highly strained and unstable relative to the Cys326-GldK disulfide. We interpret the need for NEM treatment in our experimental analysis of disulfide bonding patterns as reflecting the fact that SprF and GldK will irreversibly separate once solubilized in SDS if their disulfide linkage is broken. Consequently, even very low levels of attack on the disulfide bond between SprF and GldK by SprF Cys324 will, over time, remove the SprF-GldK adduct from the sample. By contrast, in the cellular context, SprF and GldK are kept adjacent, allowing their disulfide bond to be reformed if it is cleaved by SprF Cys324.

Both cysteine residues in the C-tail of SprF are important for gliding (Fig. 1c). This behavior is readily explained for Cys326 because this residue is required to form the disulfide bond with GldK. However, this disulfide bond is still formed when Cys324 is absent, raising the

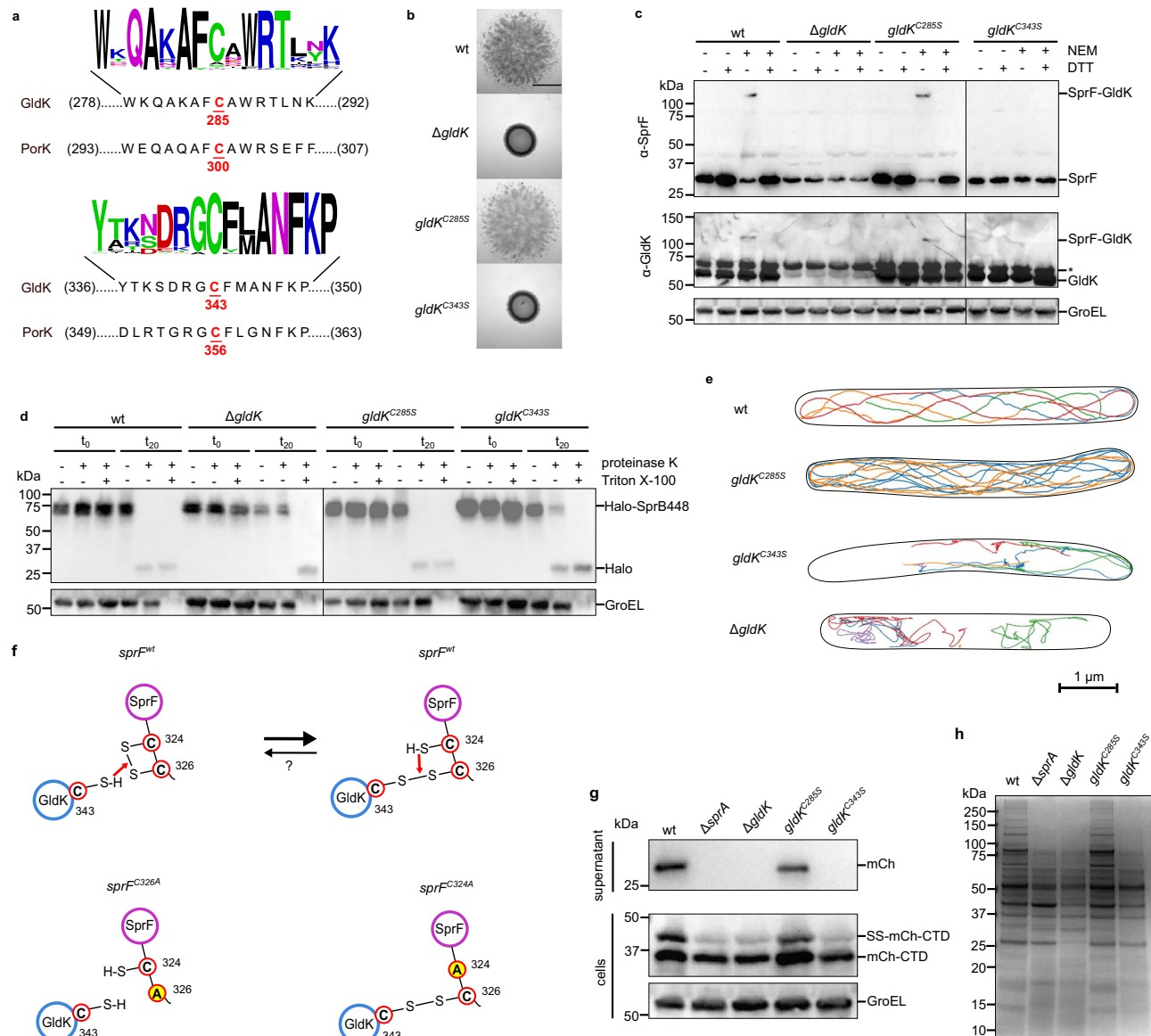

**Fig. 3 | Identification and functional characterization of the SprF-crosslinking residue in GldK. a** The sequences of the two cysteine-containing polypeptide regions of *F. johnsoniae* GldK and the corresponding regions of *P. gingivalis* PorK. Above is a Weblogo display of the sequence conservation for sequences with >40% whole protein identity to GldK. **b, c** Identification of the GldK cysteine that forms the disulfide linkage with SprF. **b** Spreading (gliding) morphology on agar. Scale bar, 5 mm. **c**, Strains with the indicated mutations were subject to whole-cell immunoblotting with the specified antisera. Where indicated, the cells were pre-treated with NEM. At the SDS-PAGE step, samples were run under either non-reducing (- DTT) or reducing (+ DTT) conditions. GroEL serves as a loading control. *, non-specific background band. **d** Cell surface exposure of the Type B T9SS substrate SprB assessed by protease protection. Strains expressing a HaloTag fusion between the signal sequence and the final 448 amino acids of SprB (Halo-SprB448) in the indicated genetic backgrounds were incubated with Proteinase K and the detergent Triton X-100 (to permeabilize the cells) as indicated. Reactions were stopped immediately ($t_0$) or after 20 min ($t_{20}$) and analysed by immunoblotting with HaloTag antibodies. The cytoplasmic protein GroEL serves as both a loading and a cell integrity control. **e** Exemplar trajectories of single fluorophore-labeled HaloTag-SprB adhesin molecules in the indicated backgrounds. The trajectories link the single-

molecule localizations in successive frames recorded for agarose-immobilized cells and are 2D projections of what is a 3D process. Each trajectory within a cell is shown in a different color, and the segmented outline of the cell is shown in black. For the $gldK^{C343S}$ background a cell was chosen to illustrate the observed range of SprB mobility behaviors. See also Supplementary Movie 2. **f** Schematic showing the proposed reactions between the cysteine residues of SprF and GldK (top) and how this is altered in single cysteine SprF variants (bottom). The pink arrows represent attacks by a free cysteine residue on an adjacent disulfide bond. Whether disulfide bond formation between SprF and GldK is reversible under physiological conditions is unknown (indicated by?). **g, h** GldK Cys343 is required for Type 9 protein secretion. **g** Analysis of the secretion of a model Type A substrate protein comprising a fusion between a signal sequence (SS), mCherry (mCh), and a T9SS-targeting C-terminal domain (CTD). Cells were separated into cell and supernatant fractions and analyzed by anti-mCherry immunoblotting. The successively processed forms of the fusion proteins are indicated to the right of the blots. GroEL serves as a loading control. **h** Secretome analysis. Culture supernatants were separated by SDS-PAGE and stained with Coomassie Blue. **b–e**, **g**, **h** Similar results were obtained for three biological repeats. wt, wild type. Δ*sprA*, a strain missing the T9SS translocon.

question of why this residue is needed for the cell to move. It is possible that the small proportion of SprF molecules that are crosslinked to GldK in this background (Fig. 2a) is insufficient to sustain gliding. It is also possible that the gliding mechanism requires the crosslink

between Cys326 and GldK to be dynamically reversible, but that this is no longer possible without Cys324 to attack the intra-protein disulfide bond (Fig. 3f). In an attempt to gain insight into these possibilities we examined the effect of the single SprF cysteine substitutions on the

mobility of fluorophore-labeled adhesins (Fig. 1e). As expected, removal of Cys326 phenocopies the behavior of other backgrounds ($sprF^{C324A,C326A}$, $gldK^{C343S}$) that cannot form the SprF-GldK disulfide bond (Figs. 1e and 3e). By contrast, removal of Cys324 produces a mix of behaviors in which some adhesins perform sustained helical movements (like cells with wild-type SprF) whilst others only move helically for short periods at best (as in strains that cannot disulfide bond to GldK)(Fig. 1e), consistent with the observation that only a proportion of the SprF molecules are disulfide-bonded to the gliding track in these cells (Fig. 2a). If, as we hypothesize, the SprF-GldK disulfide links in this strain cannot readily be reversed, then the fact that some adhesins (those we have interpreted as associating with SprF molecules that are disulfide-linked to GldK) are able to sustain continuous helical movement for multiple circuits of the cell implies that the ability to break the disulfide linkage is not a requirement for normal adhesin motion. This conclusion is strengthened by the observation that the movement of another protein that moves on the gliding track (SprC)[18] is unimpaired in the $sprF^{C324A}$ background (Fig. 1e). Thus, locking some SprF molecules to GldK does not impede the movement of the track network. In conclusion, these imaging experiments support the idea that the inability of the $sprF$ Cys324 mutant to glide is due to the low proportion of SprF molecules that are involved in disulfide bonds with GldK.

## GldK Cys343 is required for Type 9 protein export

Our findings demonstrate that Cys343 in GldK is essential for gliding motility, functioning through the formation of a disulfide bond that attaches the SprF-SprB adhesin complex to the gliding track. Remarkably, this cysteine residue is conserved in the homologous PorK protein of *Porphyromonas gingivalis* even though this organism is non-motile and does not utilize gliding adhesins (Fig. 3a). Indeed, this cysteine residue is completely invariant across GldK/PorK proteins. This strongly suggests that this conserved cysteine has a role(s) beyond anchoring adhesin carrier proteins. More specifically, since PorK is part of the *P. gingivalis* T9SS[29], it suggests that the conserved cysteine is involved in Type 9 protein export.

To test this hypothesis, we asked whether GldK Cys343 is required for Type 9 protein secretion in *F. johnsoniae*. We found that removal of this residue completely abrogated the export of a model T9SS Type A substrate protein (Fig. 3g). It also phenocopied the effects of removing GldK or the T9SS translocon SprA on the secreted proteome, which is dominated by Type A T9SS-secreted proteins[10] (Fig. 3h). By contrast, and as noted previously above, removal of Cys343 reduces rather than completely blocks export of the Type B T9SS substrate SprB (Fig. 3d). Thus, these experiments confirm the hypothesis that GldK Cys343 is involved in Type 9 protein export in *F. johnsoniae*, in addition to gliding motility, and that it is essential for the secretion of Type A substrates. Control experiments using SprF variants lacking the twin-cysteine motif confirm that the requirement for GldK Cys343 in Type 9 transport is independent of the disulfide bond that GldK Cys343 makes with SprF (Supplementary Fig. 5).

## GldK Cys343 forms a disulfide bond with the T9SS component PorG

The role of GldK Cys343 in gliding motility is to form a disulfide bond with the partner protein SprF. We, therefore, considered it highly likely that its function in Type 9 protein secretion is also to form a disulfide bond with a partner protein(s), in this case presumably a T9SS component. Indeed, a possible additional disulfide-bonded GldK adduct could be distinguished after NEM treatment of cells as a faint band in some high exposure GldK immunoblots. This is most clearly seen in the example shown in Supplementary Fig. 6.

Analysis of the sequences of known T9SS components identified the protein PorG as the most likely candidate to partner GldK. *F. johnsoniae* PorG (Fjoh_1692, hereafter $PorG_{Fj}$) is predicted to be an 8-stranded OM protein in which the periplasmic C-tail contains the

same vicinal Cys-Xaa-Cys motif used by SprF to form a disulfide bond with GldK (Fig. 4a). The presence of cysteine residues in the PorG tail is a conserved feature across the T9SS-containing Bacteroidota, although the number, spacing, and positioning of the residues varies (Fig. 4a). For example, *P. gingivalis* PorG (PG0189 in strain W83, hereafter $PorG_{Pg}$) contains a single cysteine that aligns with the second cysteine in the *F. johnsoniae* $PorG_{Fj}$ motif (Fig. 4a). $PorG_{Pg}$ was identified as an essential T9SS component in *P. gingivalis*[38,39] but its mechanistic role is unknown. Importantly, $PorG_{Pg}$ has been reported to co-purify with the *P. gingivalis* PorKN Hub Complex[33], an observation that is consistent with the hypothesis that PorG is a disulfide-linked partner of GldK/PorK.

We constructed a *porG* deletion mutant in *F. johnsoniae* and used this to confirm that PorG is part of the T9SS in this organism. Specifically, PorG is essential for the secretion of both Type A substrates and the Type B substrate SprB (Fig. 4b–d). The $porG_{Fj}$ deletion mutant was also unable to glide, consistent with its inability to export the SprB gliding adhesin (Fig. 4e and Supplementary Fig. 7a). Control experiments confirmed that the absence of $PorG_{Fj}$ did not affect cellular levels of the key T9SS and gliding components SprA (T9SS translocon), GldK, or SprF (Supplementary Fig. 7b).

We next tested whether the C-tail $Cys_{225}$-Xaa-$Cys_{227}$ motif was important for *F. johnsoniae* PorG function. Type A protein export and gliding motility were fully blocked in the absence of the cysteine pair (Fig. 4b, d, e). However, partial export of the Type B substrate SprB still occurred (Fig. 4c). This is in contrast to the phenotype of a full $PorG_{Fj}$ deletion (Fig. 4c), but equivalent to the phenotype obtained after deleting the GldK Cys343 to which we propose $PorG_{Fj}$ binds (Fig. 3d). Removal of Cys225 alone was sufficient to almost completely block export of a model Type A substrate, whereas removal of Cys227 had no substantive effect on this process (Fig. 4b), indicating that Cys225 is the more important of the two cysteine residues. Export of the Type B substrate SprB and gliding motility were either not (Cys227), or only slightly (Cys225), sensitive to the loss of the individual cysteine residues (Fig. 4d, e and Supplementary Fig. 7a). Overall, the phenotypic effects of Cys225 removal are more severe than those of Cys227 loss.

Having established that the $PorG_{Fj}$ cysteine motif is required for $PorG_{Fj}$ function we next investigated whether this motif was involved in forming the predicted interaction with GldK Cys343. In order to detect $PorG_{Fj}$ by immunoblotting we introduced an ALFA-tag into Loop 3 of $PorG_{Fj}$. This epitope tag did not affect the expression, stability, or functionality of $PorG_{Fj}$, and epitope-tagged cysteine motif variants showed the same patterns of functional defects as the untagged proteins (Supplementary Fig. 7c–e). Treatment of cells with NEM trapped a proportion of the $PorG_{Fj}$ molecules in a disulfide-linked adduct (Fig. 4f) of the same size as the additional, non-SprF, adduct of GldK detected in Supplementary Fig. 6. This adduct was absent in a *gldK* deletion mutant (Fig. 4f), confirming that GldK is the partner protein to which $PorG_{Fj}$ is disulfide-linked. Removal of GldK Cys343, but not Cys285, also blocked formation of the $PorG_{Fj}$ adduct, confirming that GldK uses the same cysteine residue to form disulfide crosslinks with both $PorG_{Fj}$ and SprF (Fig. 4f). Formation of the $PorG_{Fj}$-GldK adduct required the presence of the twin-cysteine motif in $PorG_{Fj}$ demonstrating the involvement of these residues in the disulfide linkage (Fig. 4f). The conclusion that GldK is linked covalently to PorG through a disulfide-linkage was verified by co-purification of the two proteins under denaturing conditions (Fig. 4g, Supplementary Fig. 7f).

When the $PorG_{Fj}$ cysteines are individually removed, the PorG-GldK adduct is still observed (Fig. 4f), indicating that either of the cysteines can form a bond with GldK. Nevertheless, in both cases, the levels of the $PorG_{Fj}$-GldK adduct are lower than in the wild type, with the larger effect of Cys225 removal suggesting that Cys225 is preferentially involved in forming the inter-subunit disulfide bond, in agreement with the greater functional importance of this cysteine. Removal of either of the $PorG_{Fj}$ cysteine residues results in the $PorG_{Fj}$-GldK adduct being detected even in the absence of NEM treatment

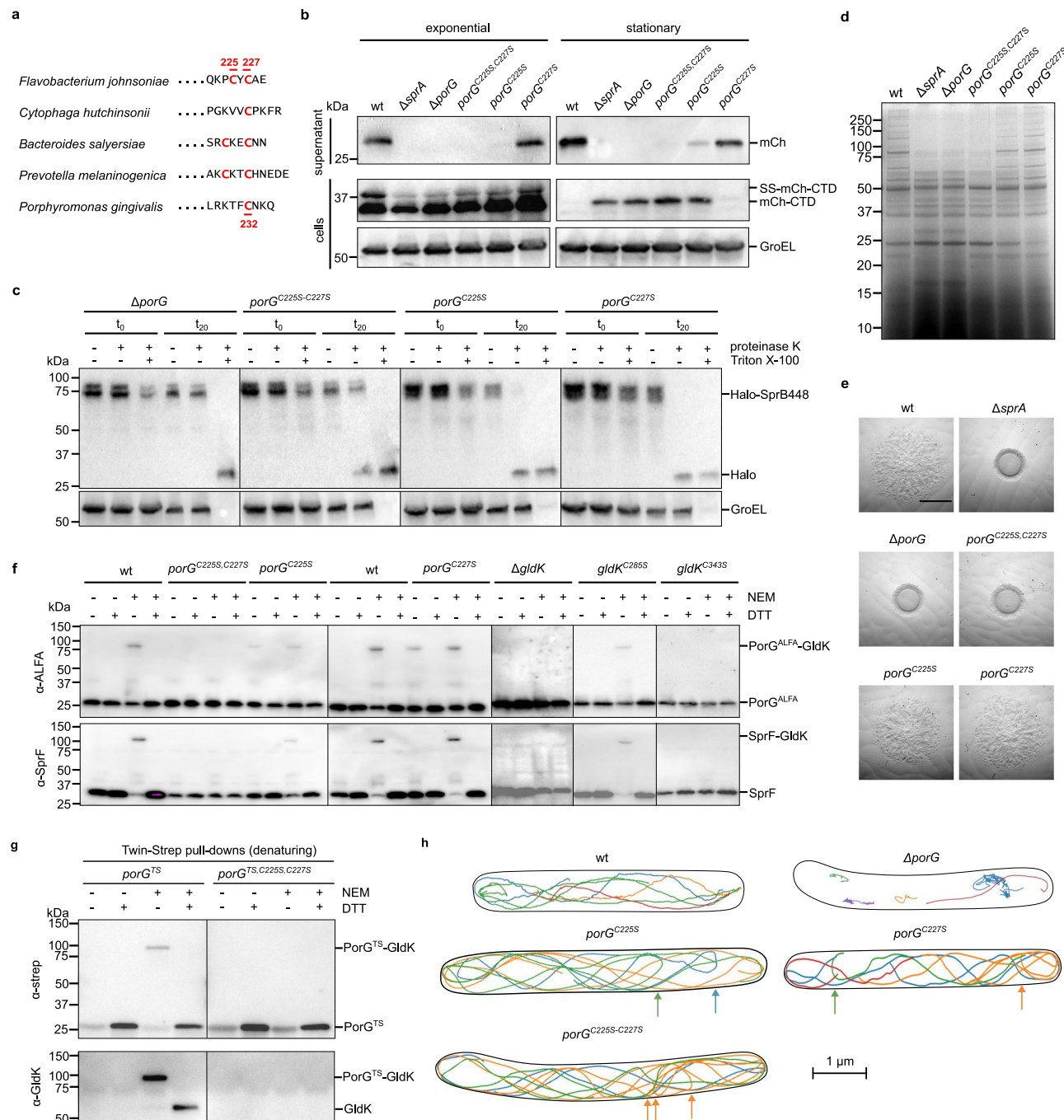

(Fig. 4f), suggesting that in these variants the disulfide bond with GldK is stabilized because it cannot be attacked by the missing PorG$_{Fj}$ cysteine. Disulfide bond formation between PorG and GldK does not require T9SS activity (Supplementary Fig. 7g).

Overall, the disulfide-bonding behaviors of the twin-cysteine motifs of PorG$_{Fj}$ and SprF are comparable, although with PorG$_{Fj}$ there is less differentiation between the roles of the two cysteines. Consequently, we interpret the behavior of the PorG$_{Fj}$ cysteines in an analogous way to the SprF motif (Fig. 3f). Specifically, we envisage that an intramolecular disulfide bond is formed between PorG$_{Fj}$ Cys225 and Cys227 during PorG$_{Fj}$ biogenesis, that this bond is subsequently attacked by GldK Cys343 to form a disulfide linkage with PorG$_{Fj}$ (preferentially involving Cys225). In the absence of either one of the PorG$_{Fj}$ cysteines, formation of the disulfide bond with GldK relies on adventitious oxidants and is inefficient.

We then examined the effects of altering the PorG$_{Fj}$ twin-cysteine motif on the motions of fluorophore-tagged SprB adhesin molecules. Adhesins still moved in continuous helical trajectories when PorG$_{Fj}$ lacked either one or both cysteine residues (Fig. 4h and Supplementary Movie 3) showing that adhesin movement on the tracks does not require the disulfide link between PorG$_{Fj}$ and GldK. Thus, the gliding defects in some of these mutants are most likely to be due to their reduced levels of adhesin export (Fig. 4c, also evident by the larger proportion of rapidly diffusing non-exported molecules in the periplasm of these strains in Supplementary Movie 3) rather than loss of adhesin function. An appreciable fraction of the adhesin trajectories in all three *porG$_{Fj}$* cysteine mutants showed mid-cell reversals rather than traveling fully to the next pole. Examples are shown in Fig. 4h and Supplementary Movie 4. This behavior is very rare in wild-type cells suggesting that track morphology is perturbed in the cysteine mutants

**Fig. 4 | *F. johnsoniae* PorG forms a disulfide bond with GldK C343 that is important for Type 9 protein transport and gliding motility. a** The periplasmic C-tails of PorG proteins contain cysteine residues. Sequence alignment of representative PorG C-tails from the indicated Bacteroidota species with cysteine residues highlighted. *F. johnsoniae* and *C. hutchinsonii* are gliding bacteria, others are non-gliding species. **b–e** The Cys residues of *F. johnsoniae* PorG are required for gliding motility and Type 9 protein transport. wt, wild type. Δ*sprA*, strain missing the T9SS translocon. **b** Analysis of the secretion of a model Type A substrate protein comprising a fusion between a signal sequence (SS), mCherry (mCh), and a T9SS-targeting C-terminal domain (CTD). Cells were grown to exponential or stationary phase as indicated, separated into cell and supernatant fractions, and analyzed by anti-mCherry immunoblotting. The successively processed forms of the fusion proteins are indicated to the right of the blots. GroEL serves as a loading control. **c** Cell surface exposure of the Type B substrate SprB assessed by protease protection. The specified strains expressing a fusion between HaloTag and the final 448 amino acids of SprB (Halo-SprB$_{448}$) were incubated with Proteinase K and the detergent Triton X-100 (to permeabilize the cells) as indicated. Reactions were stopped immediately (t$_0$) or after 20 min (t$_{20}$) and analysed by immunoblotting with Halotag antibodies. The cytoplasmic protein GroEL serves as a both a loading and cell integrity control. **d** Secretome analysis. The supernatants of cultures grown to stationary phase were separated by SDS-PAGE and stained with Coomassie Blue.

**e** Spreading (gliding) morphology on agar after 24 h growth. Scale bar, 5 mm. See Supplementary Fig. 7a for data at 6 h growth. **f** PorG is disulfide-bonded to GldK C343. All strains express PorG epitope-tagged by insertion of ALFA-tag in loop 3. Strains were subject to whole cell immunoblotting with the indicated antibodies. Where indicated the cells were pre-treated with NEM. At the SDS-PAGE step samples were run under either non-reducing (- DTT) or reducing ( + DTT) conditions. **g** GldK co-purifies with PorG through a disulfide linkage. Twin-Strep-tagged PorG (PorG$^{TS}$) variants from whole cells pre-treated or not with NEM, were isolated under denaturing conditions using Streptactin-coated beads. The bead-bound proteins were subject to immunoblotting with the specified antisera under either non-reducing (- DTT) or reducing ( + DTT) conditions. **h** Exemplar trajectories of single fluorophore-labeled HaloTag-SprB adhesin molecules in the indicated backgrounds. The trajectories link the single-molecule localizations in successive frames recorded for agarose-immobilized cells and are 2D projections of what is a 3D process. Each trajectory within a cell is shown in a different color, and the segmented outline of the cell is shown in black. Un-transported periplasmic SprB molecules have been edited out for clarity. Arrows indicate examples of mid-cell reversals of the SprB trajectories. See also Supplementary Movies 3 and 4. **b–h** Similar results were obtained from 2 (**g**) or 3 (**b–f, h**) biological repeats. wt, wild type.

and that PorG$_{Fj}$ may have a role in determining the architecture of the gliding tracks.

As noted above, the PorG$_{Fj}$ cysteine motif is not rigidly conserved. For example, the PorG protein of *P. gingivalis* is missing the cysteine residue that in *F. johnsoniae* PorG forms the disulfide bond with GldK (Fig. 4a), raising the question as to whether the remaining cysteine residue (Cys232) is required to support Type 9 function. We found that substituting this residue in PorG$_{Pg}$ abolished Type 9 export in *P. gingivalis*, as judged by the loss of colony pigmentation that is characteristic of T9SS-defective strains grown on blood agar[40] (Fig. 5a, b). Thus, the single cysteine motif of *P. gingivalis* PorG is still an essential requirement for Type 9 protein secretion. Generalizing this observation implies that all PorG C-tail cysteine motifs are involved in PorG function, but that there is flexibility in the exact complement and arrangement of the cysteine residues.

### Structure of the *P. gingivalis* PorKN Hub complex

We have shown here that the *F. johnsoniae* protein GldK and its *P. gingivalis* equivalent PorK are disulfide-linked to partner proteins in the OM. Whilst PorK is known to be a component of the *P. gingivalis* Hub complex, our data indicate that the equivalent GldK protein is part of the mobile *F. johnsoniae* gliding track. This implies that the Hub complex and gliding track are related structures, and thus that both are mobile tracks. From this standpoint, the Hub complex is a miniature circular analog of the longer gliding track.

We sought to understand the structure of these Bacteroidota mobile tracks and to characterize the structural context of the disulfide link between GldK/PorK and their partner OM proteins. Since we and others[20] have been unsuccessful in isolating the gliding tracks from *F. johnsoniae* we concentrated our efforts on the *P. gingivalis* PorKN complex which has previously been characterized at low ( > 40 Å) resolution by cryo-EM[33]. We were able to improve the structure of the ~3 MDa PorKN ring to near-atomic resolution (2.4 Å) (Fig. 5c, Supplementary Figs. 8 and 9a, Supplementary Table 1). The PorK subunit is structurally defined from Gly32 to Phe476 and thus missing the 8 N-terminal residues and 15 C-terminal residues of the mature protein. PorN is defined from Thr47 to Trp299 and so is missing the 22 N-terminal residues and 62 C-terminal residues of the mature protein (Fig. 5d). Two glycosylation sites are present on each subunit, providing glycosylation on both the outer (PorK-linked) and inner (PorN-linked) faces of the ring (Fig. 5d).

The PorKN ring is composed of multiple side-by-side PorKN heterodimer units (Fig. 5c), consistent with earlier biochemical evidence

for an equimolar PorK:PorN stoichiometry[33]. The actual number of repeats is not fixed, ranging from 31 to 35 PorKN heterodimers, but is most frequently 33 (Fig. 5c), in agreement with the earlier lower-resolution study of the complex which reported 32 to 36 repeating units[33]. Such variation in subunit stoichiometry is commonly observed for ring complexes made up from large numbers of repeating subunits[41,42], reflecting the small differences in stability of oligomers of closely similar size.

The overall shape of the PorKN complex is consistent with earlier, lower-resolution cryoEM and in situ cryoET studies of the complex[33–35], and our high-resolution structure is readily fitted into the published lower-resolution envelopes for the complex (Fig. 5e, Supplementary Movie 5). The wall of the ring is angled inwards and is almost triangular in cross-section, with the outer face more sloped than the inner (Fig. 5f). The 33-mer ring has an outer diameter of 540 Å and an inner diameter of 380 Å (Fig. 5c). Published cryoET images (and the fit of our model to these data, Fig. 5e) suggest that the face of the ring formed by the base of the triangle is positioned next to the OM[34,35]. Consistent with this assignment, the detergent micelle in our structure lies above this face of the ring and is appropriately positioned to interact with the lipidated N-termini of the PorK subunits (Fig. 5c). This membrane-facing side of the ring is noticeably flat (Fig. 5c) and not highly charged, as would be appropriate for a surface that moves across the face of the membrane. PorK and PorN form distinct layers of the ring (Fig. 5c), as previously speculated[33]. The positions of the PorN subunits are somewhat offset relative to PorK, resulting in each PorN subunit contacting two PorK subunits and vice versa. The contacts between all neighboring subunits are tightly packed and very extensive (Supplementary Movie 6). The interactions between the well-folded subunit cores are augmented by a long N-terminal arm extended by each PorK subunit across the clockwise-adjacent (as viewed from the OM) PorK molecule, and shorter arms at the termini of the PorN subunits which contact adjacent PorK molecules (Fig. 5g). These extensive contacts bury more than ~5,500 Å$^2$ of surface for every PorKN pair incorporated into the ring (Fig. 5h). This large interaction interface indicates a very stable complex in which the subunit interactions are rigidly fixed. Likewise, the bulk of the individual subunits are compactly folded, implying they are rigid structures, and there are no indications of conformational flexibility anywhere within the ring in the cryoEM data. These observations are difficult to reconcile with earlier interpretations of cryoET data as showing a 40° variation in PorKN angle driven by the PorLM motors[34].

The core of the PorK subunit has an extensively elaborated sulfatase-modifying enzyme (SME) fold[43], as previously noted[30], and

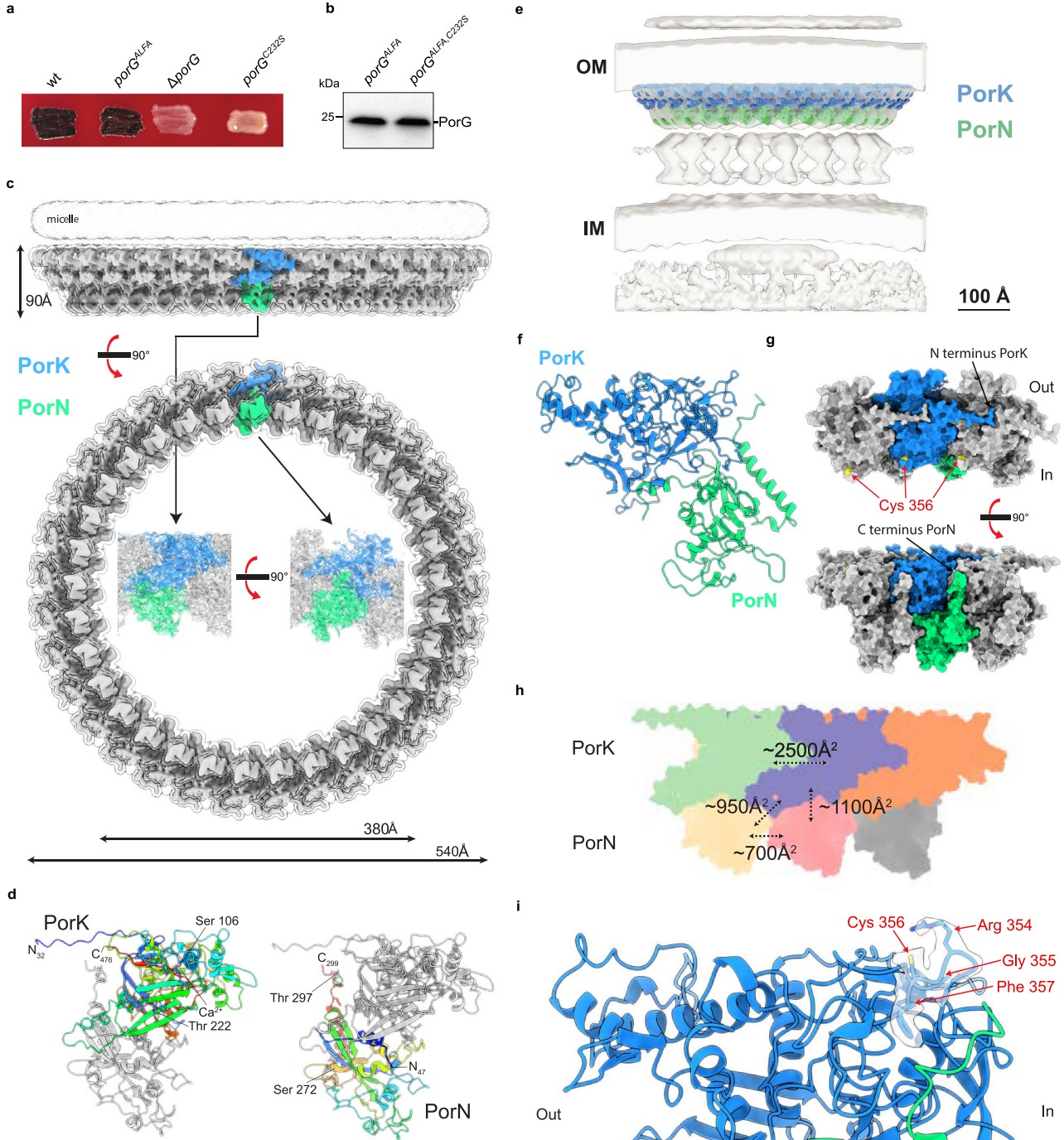

**Fig. 5 | Importance of the cysteine residue of *P. gingivalis* PorG and structure of the PorKN Hub complex. a, b** *P. gingivalis* PorG Cys232 is required for T9SS activity. Similar results were obtained from 3 biological repeats. wt, wild type. *porGᴬᴸᶠᴬ*, codes for PorG with an ALFA-tag epitope. **a** Strains containing the indicated *porG* alleles were grown on blood agar. **b** Equivalent strains constructed in a gingipain-deficient background were analysed by whole cell immunoblotting. **c** CryoEM volume for a 33-mer *P. gingivalis* PorKN complex at 4.0 Å resolution viewed from the side and from the periplasm. The inset shows the structure of the repeating PorKN unit obtained at higher (2.4 Å) resolution. The structure is depicted at a high contour level (solid representation with a single PorKN unit shown in color) and at a low contour level (transparent envelope) to visualize the detergent micelle, which approximates the position of the OM. **d** Structures of PorK and PorN in cartoon representation and rainbows colored from N- (blue) to C-terminus (red), viewed from the inner rim of the PorKN ring and with the OM at the Top. The bound Ca²⁺ ion is shown in atomic sphere representation. Glycosy-lated serine and threonine residues are labelled. **e** The PorKN structure from **c** (colored density) placed into the cryoET volume of the complex resolved pre-viously in intact cells[34] (EMD-24227). See also Supplementary Movie 5. **f** View of the PorKN heterodimer unit in cartoons representation, OM at top, illustrating the triangular cross-section of the PorKN ring. **g** Space-filling model of a PorKN pair and its immediately neighboring pairs viewed from the OM (Top) and from the outer rim of the PorKN ring with the OM above (Bottom). The position of the PorG-reactive Cys 356 on PorK is shown (yellow). See also Supplementary Movie 6. **h** Schematic summarizing the areas buried by the different inter-subunit interfaces of the PorKN complex. Protomers are oriented as in (**d**). **i** Structure of the external loop bearing the PorG-reactive cysteine 356 of PorK with density for the motif overlaid.

contains the canonical disulfide bond and bound metal atom (that we identify as a calcium ion based on the observed co-ordination, Supplementary Fig. 9b)[43] that likely structurally stabilize the fold. The extensions to the SME core found in PorK are involved in extensive interactions with neighboring subunits, adding to the stability of the ring (Fig. 5g).

The PorN subunits have a well-folded core as seen in an earlier crystal structure of this part of the subunit (RMSD 0.7 Å across 169 residues)[44]. However, both the N and C-termini become ordered in assembling the ring, with the N-terminus packing against PorK and the C-terminus packing on the inner face of the PorN portion of the ring (Fig. 5f, g).

The PorG-reactive cysteine residue Cys356 is exposed on the OM-facing side of PorK, congruent with its role in interacting with the OM-embedded PorG, and is located at the inner rim of the PorKN ring (Figs. 5g, i and 6a, b). Cys356 is found in a conserved Arg-Gly-Cys-Phe motif (Fig. 3a) with the arginine presented at the apex of a small surface loop that projects away from the protein core and is stabilized by intra-loop backbone hydrogen bonds (Fig. 5i). Within this loop the side chain of Arg354 sits above Cys356 partially screening it from

interacting partners although the cysteine side chain remains highly solvent exposed.

We used AF3 to model the interaction between PorG and PorKN. AF3 recapitulated the structure of PorKN well (RMSD 2.8 Å overall for a 3:3 heterocomplex, Supplementary Fig. 9c and Supplementary Data 1) and demonstrated that interactions between PorG and the ring are likely limited to the linking disulfide bond and interactions from residues immediately surrounding the disulfide in the unstructured tail of PorG (Supplementary Fig. 9d and Supplementary Data 2). In particular, no consistent or high-confidence interactions were predicted between the base of the PorG barrel and the upper surface of the ring. We also used AF3 to predict interactions between the PorM dimer arm of the PorLM motor that energizes the PorKN ring (Supplementary Fig. 9e and Supplementary Data 3). The tip of the PorM dimer is predicted to lie within the cleft formed between neighboring copies of PorN at the periplasmic face of the ring, with the majority of the contacts coming from one copy of the D4 domain of PorM. Although the surface buried in this interaction is not substantially conserved and the confidence metrics for this mode of interaction weak, this may reflect the dynamic nature of the interaction between PorM and the PorKN ring. Previous biochemical and genetic interaction data had suggested that PorM interacts with both PorN and PorK[44–46] but only PorN contacts are supported by our modeling.

While this work was in review the structure of the *P. gingivalis* Hub complex was also reported by two other groups[47,48].

### Model for the *F. johnsoniae* gliding track structure
Having established that AF3 reliably predicts the experimental structure of the PorKN mini-track we used AF3 to predict the structure of the analogous *F. johnsoniae* gliding track. These predictions included the Hub proteins GldK and GldN, as well as the previously established gliding track component GldJ. Predictions containing equal numbers of GldK, GldN, and GldJ protomers formed a high confidence and well-packed filamentous structure (Fig. 6b, Supplementary Data 4). Within this filament, the GldKN subunits have essentially the same arrangement as the equivalent subunits in the *P. gingivalis* PorKN structure (231 core residue superimposed with an RMSD of 0.7 Å). The GldJ subunits, which are structurally related to GldK[36], mimic the organization and placement of the GldK protomers but are on the opposite side of the filament and run in the reverse direction along the filament. The result is that GldK and GldJ pack alongside each other in two antiparallel lines on the OM-proximal side of the filament, with both subunits in contact with GldN. The packing between all three types of subunits is extensive and indicative of a single stable entity consistent with the idea that the filament represents the gliding track and that GldK and GldN are components of the track. In our model, the SprF/PorG-interacting Cys343 of GldK is positioned at the center of the OM-facing surface of the filament at the interface between the lines of GldJ and GldK subunits (Fig.6b). Our model for the interactions between GldJ and GldK differs from a recent proposal based on computational docking[49].

### Discussion
This work reveals a cellular mechanism of mechanical energy transduction in which membrane proteins are linked by disulfide bonds to a moving track structure (Fig. 7a, b). To the best of our knowledge, such a use of disulfide bonds to attach client proteins to moving elements of a machine has no analogy in any other mechanical system in biology. Remarkably, we find that this mechanism of energy transduction is shared between two of the major phylum-specific systems of the Bacteroidota – gliding motility and Type 9 protein transport – revealing that these processes are mechanistically and evolutionarily linked (Fig. 7a, b).

The key observation underpinning our conclusions is the unexpected discovery that the OM-spanning protein SprF - which links the

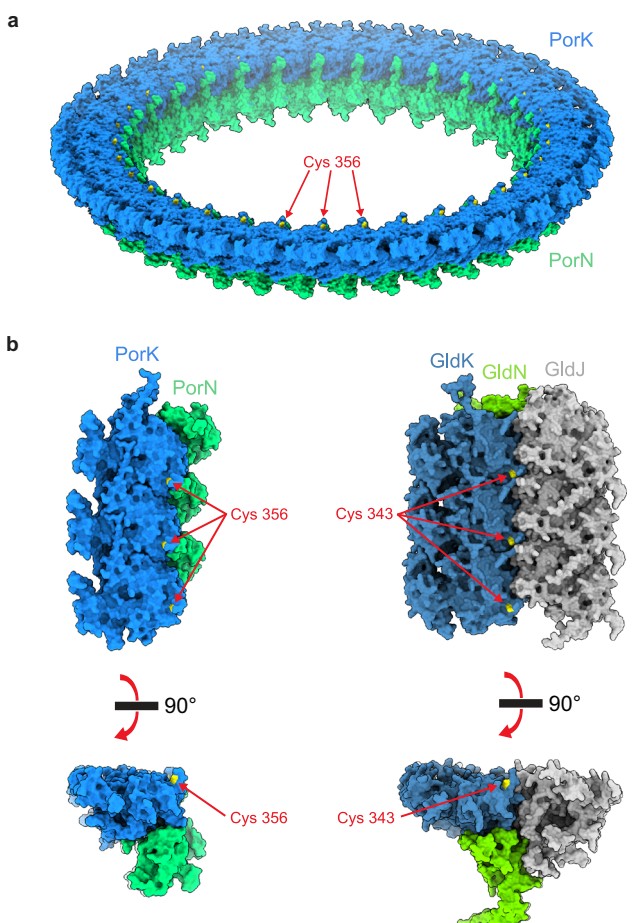

**Fig. 6 | Comparison of the structural organization of the *P. gingivalis* Hub complex and an AlphaFold model of the *F. johnsoniae* gliding track. a** The *P. gingivalis* Hub complex (PorKN ring) in spacefilling representation viewed obliquely from the OM with the PorG-interacting Cys356 residue of PorK colored yellow. **b** Comparison of the structure of the *P. gingivalis* PorKN ring oligomer (Left) with an AF3 model for the *F. johnsoniae* GldJKN track oligomer (Right). In each case three adjacent copies of the repeating unit (PorKN or GldJKN) are shown in spacefilling representation viewed from the OM (Top) or in cross-section through the oligomer with the OM at the top (Bottom). The position of the PorG/SprF-interacting cysteines (yellow) are indicated. See also Supplementary Data 4.

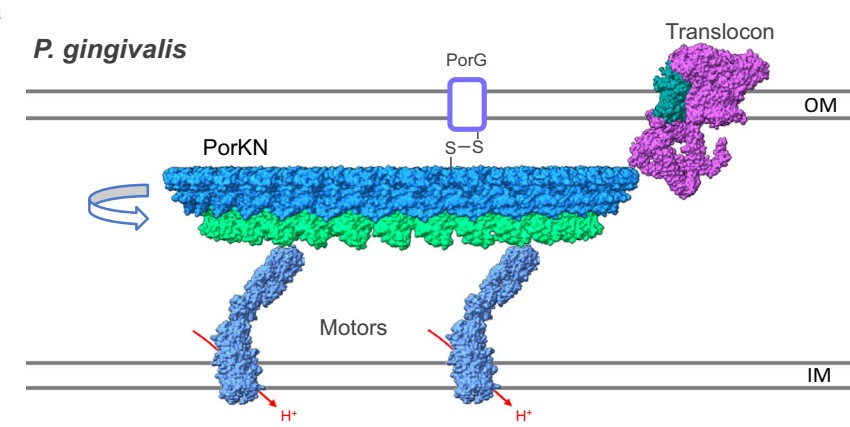

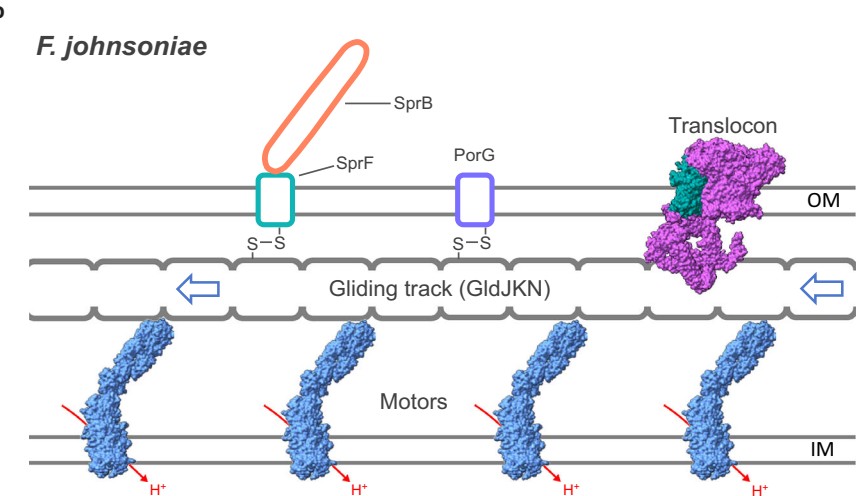

**Fig. 7 | An updated model for the connection between gliding motility and T9SS protein transport. a** Updated model for the organization of the T9SS in the paradigm non-gliding Bacteroidota *P. gingivalis* based on the findings in this work. The T9SS component PorG is disulfide-linked to the rotating PorKN Hub complex which energizes the translocon. **b** Revised model for the relationship between the gliding motility apparatus and the T9SS in the paradigm gliding Bacteroidota *F. johnsoniae* based on discoveries in this work. The adhesin carrier protein SprF and the T9SS component PorG are disulfide-linked to a moving gliding track composed of GldJ, GldK, and GldN. The translocon receives energetic input from the gliding track.

cell surface gliding adhesin SprB to the internal gliding track - is disulfide-bonded to the periplasmic lipoprotein GldK. Since almost all the SprF molecules in the cell have this disulfide linkage (Figs. 2a, b and 3c), and almost all the adhesins they carry are moving along the gliding tracks (Figs. 1e, 3e, and 4h and Supplementary Movie 1), we were able to deduce that GldK must be a component of the track. This conclusion overturns the current model for the gliding apparatus in which GldK is considered to be a component of a Hub complex that links the gliding motors to the gliding tracks[16,27,28] (Fig. 1a). In our model GldK, and its Hub partner GldN, are integral components of the gliding track (Fig. 7b), a possibility that has been briefly considered by some authors[5]. Several additional lines of evidence support our model. First, AF3 modeling predicts a high-confidence equimolar complex between the Hub proteins GldK and GldN and the known track protein GldJ. The resulting complex is stabilized by extensive tight packing between all three of the constituent protomer types, and the proteins are arranged in a repeating filament-like structure as expected of the track (Fig. 6b). The predicted complex is unlikely to represent the interfacial contact between the outside of the rotating Hub and the track filament that is envisaged in the current rack-and-pinion-model of gliding[27,28] because the structure prediction places the GldJ 'track' (rack) on the equivalent of the inner, rather than outer, face of the PorKN ring (Fig. 6b). Second, the previously observed stabilization of GldK by GldJ[50] is better interpreted in terms of GldK forming part of a GldJ-containing gliding track

than GldK being part of a separate Hub complex that does not contain GldJ. Third, a SprF variant (Cys324Ala) that we infer cannot easily reverse its attachment to GldK is still able to move helically for multiple circuits of the cell, implying that it must be attached to GldK throughout those circuits (Figs. 1e, 2a and Supplementary Movie 1). Finally, the fact that only SprF molecules that can form a disulfide link with GldK are able to mediate continuous helical movement of adhesins along the tracks (Figs. 1e and 3e and Supplementary Movie 1) is also consistent with GldK being part of the track.

Our model for the organization of the gliding track changes our understanding of how Type 9 transport is energized in both gliding and non-gliding Bacteroidota. In gliding Bacteroidota, the T9SS translocon is now inferred to be energized directly by the gliding track (Fig. 7b) rather than by a separate Hub complex (Fig. 1a), while the Hub complex of non-gliding bacteria can now be understood to be a moving track structurally analogous to the gliding track (Fig. 7a).

We envisage that the disulfide bond between SprF and the track provides a very stable linkage that allows the track to exert force on those adhesin molecules that are bound to the external substratum in order to propel the cell forward (high load). Even under the low load conditions used in our experiments, where the cells are suspended in agarose rather than pushing against a surface, the very long (~ 200 nm)[20] SprB adhesins may still require substantial force to move them through the medium surrounding the cell and, thus, still show

strong dependence on the presence of the SprF-GldK disulfide linkage. SprF is still able to interact transiently with the gliding track even in the absence of the disulfide linkage to GldK (Figs. 1e and 3e and Supplementary Movie 1). These weak interactions may be required for initial complex formation between SprF and GldK to allow subsequent disulfide bond creation.

We found that PorG, an essential component of the T9SS, is also disulfide-bonded to the track. Although PorG and SprF both bind to the same cysteine residue in GldK, it is unlikely that their binding is competitive because GldK is in large molar excess over either protein (Fig. 2a and Supplementary Fig. 6). The presence of a disulfide linkage between PorG and GldK suggests, by analogy to SprF, that PorG moves on the gliding track and that this movement results in the exertion of considerable mechanical force, necessitating covalent attachment to the track. The energy-requiring step in Type 9 transport is the ejection of carrier protein-substrate complexes from the T9SS translocon[11] and so we speculate that the role of PorG might be to transmit force from the Energy Chain to push carrier proteins out of the translocon. Intriguingly, we find that cells that fail to make a disulfide connection between PorG and GldK show apparent alterations in long-range track morphology (Fig. 4h), suggesting that PorG also has a role in track placement, perhaps by displacing other OM proteins from the track path. Export of the Type B T9SS substrate SprB is more strongly affected by removal of PorG protein than by blocking its ability to disulfide bond with GldK (Fig. 4c, h). This indicates that PorG, like SprF, possesses disulfide-independent interactions, potentially with the gliding track.

The disulfide linkages characterized here appear to be almost the only known instances of permanent inter-molecular disulfide bonds between periplasmic proteins (the OmpA proteins of the Bacteroidota *P. gingivalis* and *Tannerella forsythia* form disulfide-linked oligomers[51,52]). This probably reflects the fact that disulfide bonds are normally introduced into individual, unfolded nascent polypeptides as they enter the periplasm and are thus intramolecular in nature[53,54]. It may be for this reason that *F. johnsoniae* adopts a strategy of introducing intramolecular disulfide linkages into SprF and PorG during their biogenesis that can later be used as donor disulfides in the formation of the GldK-SprF linkage (Fig. 3f). This strategy is, however, not applicable to *P. gingivalis* PorG because this protein contains only a single cysteine residue. We note that the PorG-reactive cysteine in the *P. gingivalis* PorKN complex is highly exposed relative to the *F. johnsoniae* gliding track (Figs. 5g, i and 6a, b) which might provide sufficient accessibility to allow periplasmic thiol-oxidizing proteins to form the disulfide link with PorG in situ in a way that is not possible with the *F. johnsoniae* gliding track. It is also possible that the presence of the second cysteine in the *F. johnsoniae* PorG and SprF proteins allows the disulfide link with GldK to be reversibly broken under some circumstances (Fig. 3f), for example, to allow PorG and SprF to move between tracks or temporarily detach at track crossings, but that this is not required in *P. gingivalis* where PorG is attached to PorKN rings which operate in isolation from one another[34,35].

Based on our revised understanding of the organization of the Bacteroidota Energy Chain determined here we infer that the T9SS initially evolved as part of the Bacteroidota gliding machinery in order to place gliding adhesins onto the cell surface (Fig. 7b). This ancestral function would explain why Type 9 transport uses a very unusual mechanism in which exported proteins initially form a complex with an OM carrier protein - an appropriate strategy for the biogenesis of cell surface adhesins - rather than being directly transferred into the extracellular environment. Placing the gliding track next to the translocon in order to accept the newly transported adhesin-carrier protein complexes would have provided the translocon with a ready-made energy source to power adhesin export. We propose that the ability to glide was ancestral in the Bacteroidota but that this capability has been lost in most commensals. In these non-gliding bacteria, such as *P.*

*gingivalis*, the mobile track no longer has a gliding function but is still retained in order to power the Type 9 translocon. This more limited function allowed the cell to miniaturize the track in the form of the PorKN ring (Fig. 7a). The suggested evolutionary relationship between the gliding motility machinery and Type 9 protein transport has parallels with the relationship between the nanomachines constituting the bacterial flagellum (a motility system that exports its own components) and Type 3 Secretion Systems (which possesses only the secretion function)[55].

Our structure of the *P. gingivalis* PorKN Hub/mini-track shows that is has no substantive conformational flexibility. This suggests that force exerted on the Hub by the motor complexes cause it to rotate in place as a rigid body carrying any attached partner proteins (PorG and, by analogy to SprF, potentially carrier proteins) with it (Fig. 7a). The GldJKN filaments of *F. johnsoniae* would move linearly in a very similar manner to the PorKN ring, with the GldN subunits providing closely spaced contacts all along the filament for the gliding motors to push on (Fig. 7b). These track movements would then drag the disulfide-linked partner proteins laterally through the OM.

## Methods

### Bacterial strains and growth conditions

All strains used in this work are listed in Supplementary Data 5.

*F. johnsoniae* was routinely grown aerobically in Casitone Yeast Extract (CYE) medium[56] at 30 °C with shaking. For some physiological studies, the cells were cultured in Motility Medium (MM)[57] or PY2 medium[58].

*P. gingivalis* was grown anaerobically (5 % $CO_2$, 5 % $H_2$, 90 % $N_2$) at 37 °C in a Don Whitley Scientific A45 workstation. For liquid culture, *P. gingivalis* was grown in brain heart infusion (BHI) medium (Oxoid) supplemented with 5 µg/ml haemin, 1 µg/ml menadione, 0.5 mg/ml L-cysteine (BHIS). Culture on blood agar (BA) plates (blood agar base (Oxoid), 50 µl/ml defibrinated horse blood (HB034, TCS Biosciences Ltd), 5 µg/ml haemin, 1 µg/ml menadione, 0.5 mg/ml L-cysteine) was used to obtain single colonies. Where required, antibiotics were used at the following concentrations: 30 µg/ml erythromycin, 1 µg/ml tetracycline, 3 µg/ml carbenicillin (in BHIS liquid medium), and 40 µg/ml carbenicillin (on BA solid medium).

### Genetic constructs

Plasmids (Supplementary Data 6) were constructed by Gibson assembly[59] or Q5 site-directed mutagenesis (New England Biolabs) using the primers and target DNA in Supplementary Data 7. All *F. johnsoniae* and *P. gingivalis* mutations were constructed in the native chromosomal genes.

Chromosomal modifications were introduced into *F. johnsoniae* using the suicide vector pYT313[60] harboring the counter-selectable *sacB* gene as previously described[23]. Suicide and expression plasmids were introduced into the appropriate *F. johnsoniae* background strain by triparental mating using DH5α transformed with the plasmid to be transferred and HP101 containing the helper plasmid pRK2013 as previously described[56].

Linear DNA fragments were used for chromosomal modification in *P. gingivalis* and introduced by electroporation. Electrocompetent cells were prepared from *P. gingivalis* grown anaerobically to $OD_{600} = 0.7$. From this point manipulations were carried out outside the glove box until anaerobic conditions were re-established after electroporation. The cultures were harvested by centrifugation at 10,000 × *g* for 10 min at 4 °C, then resuspended in ice-cold electroporation buffer (10% glycerol, 1 mM $MgCl_2$) and pelleted again under the same conditions. This washing step was repeated once more before resuspending the final electrocompetent cells in 200 µl of electroporation buffer. Electroporation was performed by mixing the electrocompetent cells with 0.5–2 µg of transforming DNA in a 2 mm electroporation cuvette (165-2086, Bio-Rad) and pulsing at 2500 V

for 5 ms in a BTX ECM 399 Electroporation System (SLS). Following electroporation, the cuvette was returned to the glove box, supplemented with 1 ml deoxygenated BHIS medium, and incubated for 24 h to allow the cells to recover. If *cepA* was used as the selective marker the cultures were then diluted 1:100 in BHIS supplemented with 3 μg/mL carbenicillin and incubated until turbid (5–7 days). A dilution series was then plated on blood agar plates supplemented with 40 μg/mL carbenicillin. For other selection markers the transformed culture was plated directly onto BA supplemented with the appropriate antibiotic.

To facilitate genetic modification of *P. gingivalis* in a triple gingipain deleted background the *rgpB::ermAM-F* locus in strain EK18[61] was replaced with *cepA* using a SmaI-digested fragment of pCL54 to produce strain P54.

All plasmid constructs and chromosomal modifications were confirmed by sequencing.

### Immunoblotting and antisera used

Immunoblotting was carried out as previously described[11]. Polyclonal antibodies against GldJ and PorN were raised in rabbits against synthetic peptides. Polyclonal antibodies against SprF[14], GldK[30], and SprA[62] were provided by Mark McBride (University of Wisconsin-Milwaukee). The following commercial antisera were used: anti-HaloTag (G921A, Promega), anti-GroEL (G6532 Merck), anti-mCherry (ab167453, Abcam), anti-ALFA-Tag (N1582, Synaptic Systems GmbH), anti-Strep-tag (34850 Qiagen), anti-mouse IgG peroxidase conjugate (A4416 Merck), and anti-rabbit IgG peroxidase conjugate (31462 Pierce). Antibodies were used at the following dilutions: anti-Strep-tag 1:2000; anti-SprF 1:2500; anti-GldJ, anti-GldK, anti-PorN, anti-SprA, anti-ALFA-tag, anti-HaloTag, anti-mCherry, and all secondary antibodies, 1:3000; anti-GroEL 1:50000.

Gel and immunoblot source data are published alongside this paper as a Source Data file.

### Analytical methods

The protease protection assay for SprB export and the analysis of mCherry-CTD secretion were carried out as previously described[11].

Measurements of gliding motility on agar and analysis of the secreted proteome were performed as previously described[10].

### Whole cell analysis of disulfide-bonded protein complexes

The *F. johnsoniae* strain to be analyzed was grown overnight in 5 ml CYE medium, diluted 1:100 into 15 ml fresh CYE medium, and cultured until $OD_{600} = 0.6$. Cells were collected by centrifugation at $8000 \times g$ for 5 min, washed in PBS buffer, and resuspended in PBS buffer to $OD_{600} = 1$. The sample was aliquoted equally into two 0.5 ml fractions that were then either supplemented or not with 5 mM *N*-ethylmaleimide (E3876-5G, Merck) and incubated for 5 min in the dark on a roller at room temperature. Each sample was aliquoted equally into two 0.25 ml fractions that were then pelleted, washed in PBS, resuspended in PBS, and incubated at 100 °C for 5 min with 1x SDS-PAGE sample buffer either with or without 100 mM DTT, then analyzed by immunoblotting. For the protonophore treatment experiments, the growing culture was supplemented with 10 μM CCCP (C2759-1G, Merck) for 20 min before harvesting the cells, and 10 μM CCCP was then maintained throughout the sample preparation procedure.

### Denaturing pulldown of disulfide-bonded SprF and PorG complexes

Strains producing Twin-Strep-tagged SprF or PorG were cultured and split into two samples either treated with NEM or left untreated as in the whole cell analysis of disulfide-bonded protein complexes (above). After NEM treatment, the cells were collected by centrifugation at $8000 \times g$ for 5 min, resuspended in solubilization/denaturation buffer

(100 mM Tris HCl pH 8.0, 150 mM NaCl, 1 mM EDTA, 1% SDS) using 100 μl of buffer for cells equivalent to 1 ml at $OD_{600} = 1$, and incubated at 100 °C for 5 min. The samples were then diluted 10 times with buffer W (100 mM Tris HCl pH 8.0, 150 mM NaCl, 1 mM EDTA) to reduce the SDS concentration to 0.1%, and insoluble material removed by centrifugation at $130,000 \times g$ for 20 min. Supernatant fractions were supplemented with 10 μl ml⁻¹ BioLock (2-0205-050, IBA Lifesciences) and incubated for 10 min with constant rotation at room temperature. The samples were supplemented with 100 μl of Strep-TactinXT 4Flow High Capacity beads (2-5030-010, IBA Lifesciences) that had been pre-equilibrated in buffer WS (100 mM Tris HCl pH 8.0, 150 mM NaCl, 1 mM EDTA, 0.1% SDS) and then incubated at room temperature for 1 h with constant rotation. The beads were collected and washed 3 times with 1 ml of buffer WS. The bound proteins were eluted by resuspending the beads in 100 μl of 1x SDS sample buffer (duplicates with or without 100 mM DTT) and incubated at 100 °C for 5 min. The beads were pelleted by centrifugation and the supernatant fractions, which contain the eluted proteins, were analysed by SDS-PAGE and immuno-blotting.

### Live cell fluorescence microscopy

*F. johnsoniae* strains were cultured in MM to $OD_{600} = 0.4$ and 900 μl added to an Eppendorf tube containing 100 μl of MM pre-mixed with 1 μl of a 10 nM solution of Janelia Fluor JFX650 HaloTag Ligand (Promega) in dimethyl sulfoxide. The tubes were shaken at 30 °C for 20 min and the cells then pelleted by centrifugation at $7000 \times g$ for 3 min. The cells were washed four times by resuspension in 1 ml of PY2 medium and re-pelleting, and finally resuspended in 200 μl PY2 medium. 1 μl of this cell suspension was spotted onto agarose pads formed from 1% agarose in PY2 medium and then imaged.

All imaging data were acquired on a Nanoimager (Oxford Nanoimaging) equipped with a 640 nm 1 W DPSS laser. Optical magnification was provided by a 100× oil-immersion objective (Olympus, numerical aperture (NA) 1.4) and images were acquired using an ORCA-Flash4.0 V3 CMOS camera (Hamamatsu). All fluorescence images were collected at 15% laser power using 30 ms continuous exposure. Where applicable, data were acquired in 3D by engaging the astigmatic lens of the Nanoimager. The z-co-ordinate was calibrated by taking a z-stack in 2 nm increments in the range −1 μm to +1 μm from focus, of 0.1 μm tetraspec beads (T7279, Invitrogen) immobilized on a glass slide with poly-L-lysine.

Raw data were analysed using the Fiji plugin ThunderSTORM 1.3[63] to determine single-molecule localizations. Single molecule trajectories within cells were computed using a custom Python script that incorporated the Trackpy 0.6.2 package (https://doi.org/10.5281/zenodo.1213240), with a memory allowance of 2 frames and a search range of 3 pixels per frame. Tracks were filtered using a minimum length of 30 frames.

### Purification of PorKN complexes

PorKN complexes were purified by modifying a previously described method[33,64]. The complexes were purified from a triple gingipain-deleted mutant of *P. gingivalis* W50 (EK18). In this *P. gingivalis* strain PorK is PG_0288 and PorN is PG_0291. The strain was grown under anaerobic conditions in 10 l of BHIS broth to an $OD_{650}$ of 1. Cells were harvested by centrifugation at $15,000 \times g$ for 1 h. The pellet was resuspended in lysis buffer (50 mM TrisHCl pH 7.5, 150 mM NaCl, 0.5 M sucrose, 5 mM MgCl₂, 1% (w/v) n-Dodecyl β-D-maltoside (DDM)(Anatrace), benzonase (Sigma-Aldrich) and cOmplete™ Protease Inhibitor Cocktail (Sigma-Aldrich)) and gently mixed using a magnetic stirrer at 4 °C for 3 h. The resulting lysate was clarified by centrifugation at $10,000 \times g$ at 4 °C for 1 h. Na₂EDTA was added to the clarified supernatant to a final concentration of 10 mM, and then lysozyme powder (Sigma-Aldrich) was added to a final concentration of 0.5 mg/ml. The mixture was then incubated at 37 °C for 45 min followed by centrifugation at $210,000 \times g$ at 4 °C for 90 min. The resulting pellet was

resuspended in resuspension buffer (10 mM Tris HCl pH 7.5, 500 mM NaCl, 5 mM $Na_2EDTA$, 1% (w/v) DDM) and centrifuged at $10,000 \times g$ at 4 °C for 15 min to remove unsolubilised material. The recovered supernatant was mixed at a ratio of 1:3 with a 30% CsCl solution (30% (w/v) CsCl, 10 mM Tris HCl pH 7.5, 0.5% (w/v) DDM] and then centrifuged at $215,000 \times g$ at 20 °C for 19 h in a SW55 Ti rotor (Beckman-Colter). Fractions were sampled from the top downwards with a wide-bore pipette tip. 10 soluble fractions were isolated, followed by a loose pellet (LP) fraction containing the soluble PorKN complexes, and an insoluble pellet (P) fraction. Fractions were diluted at a ratio of 1:2 with dilution buffer (10 mM Tris HCl pH 7.5, 500 mM NaCl, 0.5% (w/v) DDM, 5 mM $Na_2EDTA$) and the PorKN complexes pelleted by centrifugation at $210,000 \times g$ for 2 h. The pellets were resuspended in sample buffer (10 mM Tris HCl pH 7.5, 500 mM NaCl, 0.5% (w/v) DDM) and successful fractionation confirmed by immunoblotting using a PorN antibody. The LP fraction (400 μL) was loaded onto a discontinuous sucrose gradient formed by layering equal volumes (400 μL) of, successively, 2 M, 1.9 M, 1.8 M, 1.7 M, 1.6 M, 1.5 M, and 1 M sucrose in dilution buffer. The gradients were centrifuged at $215,000 \times g$ at 12 °C for 17 h in a SW55 Ti rotor (Beckman-Colter). Fractions were sampled from the top downwards with a wide-bore pipette tip, 11 fractions were isolated (F1 to F11, with F1 being the upper-most fraction). Fractions were diluted at a ratio of 1:2 with dilution buffer and the PorKN complexes pelleted by centrifugation at $250,000 \times g$ for 1 h 45 min. The pellets were resuspended in sample buffer and then analysed by SDS-PAGE and by immunoblotting using a PorN antibody to identify the fractions (typically F6 and F7) containing the highest concentration of PorKN rings. These fractions were pooled and pelleted at $186,000 g$ for 2 h at 12 °C. The pellet was resuspended in 30 μl sample buffer and used to prepare cryoEM grids.

### Cryo-EM sample preparation and imaging
3 μl of the PorKN sample ($A_{280nm} = 7.2$) was applied to ultra-thin carbon-coated grids (Quantifoil 300 mesh, Au R2/1) at 4 °C in 100% humidity. The sample was adsorbed for 60 s, blotted 3 s, and plunge frozen in liquid ethane using a Vitrobot mark IV (FEI).

Movies were collected in counted mode, in Electron Event Representation (EER) format, on a CFEG-equipped Titan Krios G4 (Thermo Fisher Scientific) operating at 300 kV with a Selectris X imaging filter (Thermo Fisher Scientific) and slit width of 10 eV, at 165,000x magnification on a Falcon 4i direct detection camera (Thermo Fisher Scientific), corresponding to a calibrated pixel size of 0.732 Å. Movies were collected at a total dose of 55.0 (Supplementary Table 1), fractionated to ~ 1.0 $e^-/Å^2$ per fraction for motion correction.

### Cryo-EM data processing
Patched motion correction, CTF parameter estimation, particle picking, extraction, and initial 2D classification was performed in SIMPLE 3.0[65]. All downstream processing was carried out in cryoSPARC 4.5.3[66]. Global resolution was estimated from gold-standard Fourier shell correlations (FSCs) using the 0.143 criterion, and local resolution estimation was calculated using the 0.5 criterion.

The cryo-EM processing workflow for PorKN is outlined in Supplementary Fig. 8. Briefly, particles were downsampled from 0.732 Å/pixel to 1.464 Å/pixel and subjected to one round of reference-free 2D classification (k = 100) using a 600 Å soft circular mask. Selected particles (162,414) were further down-sampled to a pixel size of 2.928 Å and subjected to another round of 2D classification (50 classes, no mask). Particles in all tilted and side view classes were selected, together with those in top-view classes where apparent 33-fold symmetry could be counted, to generate a selection of 124,099 cleaner particles. A low-resolution volume was generated from these particles after ab initio reconstruction with D33 symmetry applied. Corresponding particles were then non-uniformly refined (applying D33 symmetry,

10 Å lowpass filter) against this volume, resulting in a Nyquist-limited 5.9 Å volume. Unmasked 3D classification of the particles was then performed against 10 classes using a resolution filter of 8 Å while applying D33 symmetry in order to remove particles from the side and tilted 2D classes that had not been visually selected as having this symmetry. Particles belonging to the most symmetric and structured volume were re-extracted to a pixel size of 1.7568 Å/pixel after re-centering using prior aligned shifts. This was followed by non-uniform refinement (D33 symmetry) against a corresponding 10 Å lowpass-filtered volume that generated an improved 4.0 Å volume. Local refinement using a rotational search extent of 5 degrees and shift extent of 3 Å was performed on D33 symmetry-expanded particles (1,097,778) in C1 against the corresponding map lowpass-filtered to 8 Å, using a soft mask covering multiple subunit copies on the ring, resulting in a 3.5 Å Nyquist-limited reconstruction. Particles and corresponding volume were then re-centred to the mask centre of mass and re-extracted using these aligned shifts to a pixel size of 1.1764 Å/pixel. These particles were subjected to local refinement using limited searches (5° angular, 3 Å shifts), resulting in a 3.1 Å volume. Local CTF refinement followed by an identical local refinement scheme further improved map quality to 2.5 Å. Particles were then re-extracted using their aligned shifts to a pixel size of 1.097 Å, followed by an additional round of C1 local refinement that generated a slightly improved 2.5 Å volume. Focused 3D classification using a mask covering three sub-units on the ring was performed against 4 classes in C1 and using a resolution filter of 3 Å. Particles belonging to the most populated class were selected and underwent local refinement (C1), generating a 2.4 Å volume from 676,608 particles.

### Model building, structure refinement, and figure preparation
Iterative model building and real-space refinement using rotamer, and Ramachandran restraints was performed in Coot 0.9.8.95[67] and Phenix 1.21[68], respectively. Validation was performed in Molprobity 4.5.2[69] within Phenix. Cryo-EM data collection, image processing and structure refinement statistics are listed in Supplementary Table 1. Figures were prepared using UCSF ChimeraX 1.9[70].

AlphaFold modeling used AlphaFold3[37]. The first-ranked model in each case is shown in the Figures.

### Statistics and reproducibility
All experiments shown in the paper are representative with similar data obtained for at least three biological repeats unless otherwise specified.

### Reporting summary
Further information on research design is available in the Nature Portfolio Reporting Summary linked to this article.

## Data availability
Cryo-EM density maps and atomic coordinates are deposited in the Electron Microscopy DataBank (EMDB) with the following accession number: EMD-48769. Atomic coordinates are deposited in the Protein Data Bank (PDB) with the following accession number: PDB 9MZU. Gel and immunoblot source data are published alongside this paper as a Source Data file. AlphaFold models generated in this study are available in Supplementary Data 1 ($PorK_3PorN_3$), Supplementary Data 2 ($PorK_3PorN_3PorG_1$), Supplementary Data 3 ($PorK_3PorN_3PorM(D3-D4)_4$), and Supplementary Data 4 ($GldJ_3GldK_3GldN_3$). Requests for materials should be addressed to BCB. Source data are provided with this paper.

## Code availability
Original code used in this study is available at https://github.com/notdrjones/Single-Particle-Tracking-for-Disulfide-Bond-Paper.

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

## Acknowledgements

We thank James Horne for producing the PorN and GldJ antisera and F. *johnsoniae* ΔgldK and ΔporG mutants, Mark McBride for providing SprF and GldK antisera, Frederick Lauber for producing the ΔsprF mutant, Kevin Foster for access to anaerobic culture and imaging facilities, and James Garnett, Mark Roberts, Joseph Aduse-Opoku, Michael Curtis, and Graham Stafford for strains, reagents, and advice on working with *P. gingivalis*. We acknowledge the use of the Oxford Micron Advanced Imaging Facility. This work was supported by European Research Council Advanced Award 833713 (BCB), Biotechnology and Biological Sciences Research Council grant BB/S007474/1 (BCB and SML), and a Medical Research Council studentship to RJ. This research was supported in part by the Intramural Research Program of the NIH.

## Author contributions

X.L. carried out all genetic and biochemical experiments, with the exception that MA purified the PorKN complex. J.C.D., C.L., and S.M.L. collected electron microscopy data and determined all structures. R.J. and X.L. carried out fluorescence microscopy experiments. X.L., B.C.B., and S.M.L. conceived the project. B.C.B. and S.M.L. secured funding. The draft manuscript was written by X.L. and B.C.B. All authors interpreted data, produced figures, and revised the manuscript.

## Competing interests

The authors declare no competing interests.
