## [Transparent Peer Review file · Nature Communications]

A shared mechanism for Bacteroidota protein transport and gliding motility

Corresponding Author: Professor Ben Berks

Version 0:

Reviewer comments:

Reviewer #1

(Remarks to the Author)

The authors present data suggesting disulfide bond formation between SprF and GldK. They extrapolate these data to strongly suggest that GldK must form the track. Overall, the data for this model is minimal, rigor is low, and further experimental validation is necessary to substantiate the proposed conclusions.

Below are detailed comments on the manuscript.

Line 22 – “structure of a representative Bacteroidota mobile track” :: This would not be accurate to describe the output. The description of tracks is based on the AF3 modeling and thus is in all essence a prediction and not a fact based on experimental evidence. This should not be called “representative” either given differences in T9SS architecture across the phylum such as the two strains used in this work- while PG principally employs T9SS for secretion, it is intricately linked to both, the secretion and the gliding motility in FJ.

Line 24 – “Our discoveries identify a novel mechanistic and evolutionary link between gliding motility and T9SS-dependent protein transport” :: It would be more accurate to indicate that this work proposes an alternative model based on Biochemical and structural evidence combined with AF-predictions.

Line 41 – the authors write ‘Power chain’ and cite ref#11. What they exactly mean by ‘power chain’ should be described here. Was this exact term used in ref#11?

Line 48: ‘present in most free living Bacteroidota’ – this is an overstatement. The citation #16 is a review where this system is briefly described along with all other bacterial motility systems. Is there a study of most free living Bacteroidota and their motility?

Line 57: citation # 24, 25 are used to support a statement on SprF – do these two articles have any experiments on SprF?

Line 55: “SprF is, in turn, thought to bind to a moving track”.. To the best of my understanding none of the cited references specifically indicate that sprF could be the entity that links sprB to the track. However, given the fact that 1. SprB protects sprF from proteolytic cleavage in intact cells 2. co-IP experiments have shown direct interactions and 3. that SprB moves over a helical ‘track’ in combination suggests the possibility of SprB-sprF-track arrangement. But there is no direct fluorescently tagged SprF known to move helically neither is there any cryo-ET or cryo-EM data in support. Since this sets the premise of the work, authors could write this clearly and cite appropriate and specific references for wider readership and be more transparent in their writing.

Line73-74 – “contacts the outer ring” - none of the cited references show this via direct experimentation – at this point, it is mostly a very plausible hypothesis.

Line 117 – “removing the cysteine pair” – authors often use the word removing when they supposedly mean substituting. A consistent use of the correct indication is encouraged to remove ambiguity.

Line 118– “Thus, the twin-cysteine motif is required for SprF to make normal interactions with the gliding track” – the finding that substitution of cysteine residues leads to loss of sprB dynamics while retaining the export does not necessarily mean that the interaction of sprF with the “track” was lost.

Even when using this claim further it is imperative that the authors experimentally demonstrate that sprF associates with track proteins. Without establishing this as an experimentally validated fact, the loss of function leading to abrogated adhesin tracking, can not be attributed to the cysteine subs.

Line 128– “Thus, SprF is still able to interact with the gliding tracks...” this partially contradicts the categorically stated inference in line 118. However, the larger issue still is the assumption that sprF interacts with track in the absence of direct experimental evidence.

Line 171 –” Thus, the Hub component GldK is the partner protein to which SprF is disulfide linked. Because SprF is in continuous movement along the gliding track, the covalent link between SprF and GldK implies that GldK is a component of the track rather than being located in a separate Hub complex as envisaged in the current gliding model”

There are several caveats to this conclusion:: 1. A successful probe of two different proteins with their cognate antibodies at the same band size after disulfide-crosslinking can not be an adequate proof that a direct interaction between these two proteins exists. Authors must confirm this interaction with more definitive methods such as mass-spec of the band-cutouts and peptide identification from them. 2. It may be assumed by some educated and confident extrapolation (as indicated earlier in the comments) that sprB-sprF-track -connection might exist, but the claim that sprF moves along the track must be experimentally demonstrated. In addition, it would perhaps be too presumptuous to claim based on the assumption that GldK is a part of “track”. That claim, in its own regard must be substantiated by experimental evidence of the highest rigor, such as labelling GldK and showing it in situ to localize along the “track” via cryo-ET.

Line 179 –”substituted providing presumptive evidence that it is Cys343 that forms the disulfide linkage with SprF...” the current experimental evidence can only support the inference that the cys343 is essential to the function of GldK and cannot support the above claim– again an analytical method such as mass spec can be useful for more definitive answer.

Line 182 – “...albeit with lower efficiency than in the wild-type strain (Fig. 3d)” the ‘lower efficiency’ claim cannot be accurately made especially given that the intensity of both of the ProtK and TX100 bands in C343S as compared to the WT is higher. In addition, there is greater variation seen in the GroEL loading control bands”. This is of course just the visual interpretation and authors can perhaps supplement quantification with densitometric analysis.

Line 192 – “...Cys343 on their mobility are equivalent to those of removing the SprF twin-cysteine motif. This is as expected if these two groups of cysteine residues are partners in the same disulfide bond”

Line 204 – “Our interpretation of this observation is that Cys324 forms an intramolecular disulfide bond with Cys326, and that this bond is then attacked by a cysteine thiol on GldK to form the SprF-GldK disulfide link” :: this sounds like a very plausible explanation. If this can not be backed by experimental evidence such as from advanced analytical methods, it may at least be helpful to cite example(s) of other proteins with twin cys motifs, where such intermolecular disulfide bonds may have been demonstrated.

Line 290– what kind of analysis? It would be a good context if the logic are explained here.

Line 331 – “This adduct was absent in a gldK deletion mutant (Fig. 4f) confirming that GldK is the partner protein to which PorGFj is disulfidelinked.” – again, same comment as before- cooccurrence of bands in presence/absence of NEM on gels is inadequate experimental evidence for this claim. A strong analytical method must confirm this finding.

Line 383– the sentence does not sound complete or has an extra word (proteins in the OM protein)?

Line 386 – to reiterate, GldK being part of the track in Flavobacterium would necessitate strong experimental evidence including but not limited to its distribution across the cell in a helical fashion as the “track”.

Line 386– “This implies that....” this is an interesting thought, especially having demonstrated and discussed (albiet with inadequate evidence) the parallel between ‘pulling’ cargo off the translocon and transporting an adhesin on the ‘track’, however this claim requires a lot more and stronger experimental evidence as indicated prior in the comments.

Line 425 – “The contacts between...” it will be helpful if this sentence was split to improve understanding.

Line 470 –”dynamic” can be interpreted in multiple ways here. Do authors mean dynamically changing interaction in the cell between the two proteins or do they mean different ways the interaction may occur in different bacteria? Some clarification could be more useful to understand the authors’ point of view.

Line 476 – “Mini-track” – As suggested in the comments before, the bonafide status of “track” or “mini-track” should be reserved until adequately backed by experimental evidence. This mini is thus far called as a ring – Porphyromonas

gingivalis does not have gliding motility – why does it need a mini track?

General comments:

The AF3-prediction that GldK is part of the 'track' - AF3 models of multiprotein complexes can provide a hypothesis and they need to be experimentally verified – I suggest the authors present their model as an alternative hypothesis and test it in future via rigorous experimentation.

Authors could mention the purpose of Triton X-100 (intact cells vs cell lysate) in either methods section or in one of the figure legends. This will improve on the accessibility of the work for all reader-levels. Particularly, not using it in the experiment of fig.1 but everywhere else - would get a better context. The cited references does not explain this either so this would be a great addition.

The figure legends should show the number of trajectories or sprB foci analyzed for TIRF experiments, and associated statistics. It would also be ok- perhaps better- to have a histogram of relevant property as additional figure panels, for eg., but not limited to, reversal frequency in different strains in fig4g (C225S)

Editorial:

The authors might want to double check the doi URL's. Some of them did not work for me.

Reference 20 – what is this? There is nothing but the name of authors, title, year and I was not able to find this article online.

(Remarks on code availability)

Reviewer #2

(Remarks to the Author)

This manuscript by Liu and colleagues explores the structure and function of the type nine secretion system (T9SS) and of the related Bacteroidota gliding motility machinery. Using genetic, biochemical, and structural studies the authors made the unexpected discovery that key outer membrane gliding (SprF) and T9SS (PorG) proteins form disulfide bonds to cysteines on GldK. In *P. gingivalis*, PorK (the homolog of GldK) is part of a ring complex that interacts with the PorLM (GldLM) motor to energize secretion. *F. johnsoniae* GldK is involved in both gliding and in T9SS-mediated secretion. The authors demonstrate that formation of the disulfide bonds are critical for proper function of the gliding and T9SS machineries. In the case of the SprF-GldK disulfide bonds, these strongly attach the outer membrane gliding motility protein SprF to GldK on the helical motility tracks that underlie the outer membrane. It was previously shown that the external motility adhesin, SprB, that is propelled along helical paths along the cell surface, interacts strongly with the outer membrane protein SprF. The results presented here provide the link between SprB and the internal track system that in turn interacts with the energizing GldLM motor in the cytoplasmic membrane to result in gliding. The story is similar for the T9SS protein PorG. The PorG disulfide bonds with GldK/PorK, and the PorG cysteines needed for this are important for proper T9SS-mediated secretion in both *P. gingivalis* and in *F. johnsoniae*.

The authors also used cryoEM to explore the structure of the *P. gingivalis* PorKN ring complexes. *P. gingivalis* is nonmotile, lacks GldJ, and appears to have a simpler GldK/PorK complex architecture than does the motile *F. johnsoniae*, where direct observation of the complex at high resolution has been difficult. The authors used the *P. gingivalis* PorN/PorK structures that they observed to model the more complex tracks that appear to be formed by *F. johnsoniae* GldJ, GldK, and GldN.

The authors present an intriguing model where the PorKN rings, which are only involved in *P. gingivalis* T9SS-mediated secretion, are a simplified version of the helical track structures formed in *F. johnsoniae*, where the proteins appear to function directly in both gliding and in secretion.

The results are convincing and are important in understanding the structure and functioning of the T9SS and the gliding motility machinery. Since both T9SS-mediated secretion and motility have been linked to virulence, and also impact the functioning of many bacteria of environmental or biotechnological relevance, they may also lead to practical applications in the future. The manuscript is well written and presents interesting, surprising, and important findings. The results greatly increase our understanding of gliding motility, by providing evidence for a direct chain of proteins linking the substratum over which the cell glides, with the gliding motor in the cytoplasmic membrane. They also provide a model for how some of the core components of the T9SS participate in both cell movement and in protein secretion. I have only minor suggestions for improvement, as listed below:

1) Is it known how SprF interacts with SprB? From previous studies this probably involves the C-terminal region of SprB, but I don't know if the exact regions of interaction are known. I am not suggesting additional experiments on this point for this paper. Just request to add information if it is already known/published.

2) p 2 line 21 "moving internal track structure which propels them through the membrane."
and p 26 line 610 "These track movements would then drag the disulfide-linked partner proteins through the OM."

"Through the membrane" may suggest to some that they are moving 'across' the membrane (from inside to out for example). Would 'along the membrane' be better? I am not sure. Choose what you think works best here.

In addition, for p 2 line 21, 'which' should be 'that'.

3) p 3 line 28

referring to Bacteroidota the authors state "They are the most abundant Gram-negative commensals in the human gut and "

Are they always the most abundant G- commensals? If yes, then no change needed. If this is not certain, then i suggest change to something like:

"They are often the most abundant Gram-negative commensals in the human gut and "

4) p 5, 6 line 96 - 97

"Substituting either or both of the (SprF) cysteine residues abolished gliding (Fig. 1c)."

Fig 1c shows colonies that either spread because of gliding motility (wild type), or that did not spread or at least spread much less (sprF mutants). It is clear that motility was affected by the mutations. What is not entirely clear is that motility was completely abolished. The authors did not examine movement of individual cells over surfaces but rather looked at the collective spreading. It is known that disruption of *F. johnsoniae* sprF or of other 'spr' genes typically results in nonspreading colonies (hence the name 'spr'), but the individual cells retain some limited ability to glide on some surfaces. This is in contrast to disruption of *gld* genes, which typically results in complete inability of cells to move/glide. The sprF mutants constructed by the authors are certainly defective in gliding, but i suspect that like other spr mutants, they retain the ability to make slight movements on some surfaces. With this in mind, 'abolished gliding' seems unnecessarily strong.

similarly, in other places in the text:

p 11 line 230

"Both cysteine residues in the C-tail of SprF are essential for gliding (Fig. 1c)"

'essential for gliding' seems too strong. Perhaps 'important for gliding' would be more accurate.

p 39, (Fig 1 title) "The cysteine residues in the periplasmic C-tail of SprF are essential for SprB movement and gliding motility"

This figure title is not accurate for the reasons described above. The figure actually demonstrates movement of some SprB molecules along the cell surface of some of the mutants that lack the sprF cysteine residues. Clearly, the movement of SprB is affected by the loss of these SprF cysteines, but 'essential for SprB movement' seems too strong, and actually incorrect.

As indicated above, there are many ways to fix these phrases, such as by indicating 'important for gliding' etc. rather than 'essential'.

Also, note that complete loss of gliding is not expected for sprF mutants. SprF is only known to interact with one adhesin, SprB. Deletion of sprB does not completely eliminate gliding. There are other adhesins that can replace SprB and function in gliding on some surfaces. Presumably they interact with the track in a similar way. Note that these suggestions do not question the importance of your results, just the accuracy of your language here.

5) p 9 line 171

"Because SprF is in continuous movement along the gliding track, "

It might be better to indicate 'Because SprF appears to be in continuous movement along the gliding track,' .

You only monitored the movement of SprB. Your results and previous results do indicate a strong interaction of SprB and SprF, so i think you are correct, but perhaps the language should be more careful here, since you did not directly observe 'continuous movement' of SprF.

6) p 14 line 301 "an observation which is consistent with the hypothesis that PorG is a disulfide-linked"

'which' should be changed to 'that'.

similarly, on p 31 line 706, "0.5ml fractions which were then either supplemented", 'which' should be changed to 'that' (or you could add comma after 'fractions' (fractions, which...), but i think 'that' makes more sense here).

7) p 52, lines 1064-1065. Extended data fig 3.

"and a T9SS-targeting C-terminal domain (CTD). whole cell and supernatant samples analyzed by anti-mCherry immunoblotting."

Needs attention. Period in wrong place? Should this all be one sentence?

8) Ref 20. Journal name?

9) Ref 37. Thesis? if so, indicate.

(Remarks on code availability)

Reviewer #3

(Remarks to the Author)

(Remarks on code availability)

Reviewer #4

(Remarks to the Author)

In this manuscript, Liu and co-workers report a biochemical and structural characterisation of the T9SS-mediated gliding mechanism in Bacteroidota. This work significantly transforms our understanding of both gliding motility, and T9SS-mediated protein transport. Specifically, it very elegantly demonstrates that the protein GldK is a component of the track, employed by both gliding and secretion machineries; and that it binds to the adhesion adapter protein SprF through disulphide bonds. This mechanism supersedes previous models, and is supported by extensive validation through structure-function analysis. Accordingly, I am highly supportive of its publication in Nature Communications (though I do wonder if it would be best suited for Nature Microbiology, considering its broad impact for our understanding of bacterial motility and secretion), and I only have a few minor comments for suggested edits.

- I think the term "new" or "novel" throughout the manuscript (in the title, the legend for Fig 6, and elsewhere) is perhaps not appropriate, and will likely be challenged by the journal guidelines anyway. So I would recommend the authors to remove these terms, and refer to "updated" and/or "refined" models as appropriate.

- I recommend that the authors include an additional supplementary video, illustrating the structure of the PorKN complex, and the various interfaces observed between these proteins. An additional video showing their fit within the low-resolution cryoET map, could also be helpful.

- The results section makes the structural analysis of the PorKN complex sound absolutely trivial, but the methods section (particularly Supplementary Figure 6 - labeled as figure 7 in the methods section!) show that it was actually rather complex. I recommend including a few additional sentences in the corresponding part of the manuscript, describing the cryo-EM data processing process. Considering the size of the complex, I would have expected major preferred-orientation issue; was this the case? If so, how was it dealt with? Similarly, the authors refer to various stoichiometries for the complex, from 31-fold to 35-fold. How were these identified? Could these be separated through classification? Additional data could be helpful here.

(Remarks on code availability)

N/A

Reviewer #1 (Remarks to the Author):

The authors present data suggesting disulfide bond formation between SprF and GldK. They extrapolate these data to strongly suggest that GldK must form the track. Overall, the data for this model is minimal, rigor is low, and further experimental validation is necessary to substantiate the proposed conclusions.

Response: We strongly refute the suggestions that the data for our model is minimal, that our experiments lack rigour, or that our conclusions require further validation. We respond to specific points made by the Reviewer below, but note that the other Reviewers have no issues with our data quality or evidence (Reviewer 2: ‘the results are convincing’; Reviewer 4: ‘is supported by extensive validation’).

Line 22 – “structure of a representative Bacterioidota mobile track” :: This would not be accurate to describe the output. The description of tracks is based on the AF3 modeling and thus is in all essence a prediction and not a fact based on experimental evidence. This should not be called “representative” either given differences in T9SS architecture across the phylum such as the two strains used in this work- while PG principally employs T9SS for secretion, it is intricately linked to both, the secretion and the gliding motility in FJ.

Response: The structure referred to is of the *P. gingivalis* PorKN complex. This is a high resolution cryoEM structure NOT an AlphaFold prediction so the Reviewer is incorrect in asserting that it is a prediction. We accept that ‘representative’ could be too broad and have changed this wording to ‘exemplar’.

Line 23.

Line 24 – “Our discoveries identify a novel mechanistic and evolutionary link between gliding motility and T9SS-dependent protein transport ” :: It would be more accurate to indicate that this work proposes an alternative model based on Biochemical and structural evidence combined with AF-predictions.

Response: This sentence refers to our identification a disulfide linkage mechanism and its use by both the gliding motility apparatus and the T9SS. This is a discovery and not a model and our formulation is justified (note that we have removed the term ‘novel’ here in response to Reviewer 3).

Our proposal for the structural organisation of the gliding track is indeed a model (though heavily supported by experimental evidence as well as close structural homology to our structure of the PorKN complex, and so is not ‘just’ an AF model), but that is not what this sentence primarily refers to.

We note also that our model is not an ‘alternative’ model. Previously published models are inconsistent with our new data and so these models are incorrect and not alternatives to our model.

Line 41 – the authors write ‘Power chain’ and cite ref#11. What they exactly mean by ‘power chain’ should be described here. Was this exact term used in ref#11?

Response: The nature of the Power Chain is not information that is required at line 41. Where it becomes necessary to describe this concept (line 64 in the original manuscript) we do so: ‘In current models the Power Chain is composed of...’.

The Reviewer is correct in pointing out that the term ‘energy chain’ rather than ‘Power Chain’ is used in reference 11. For consistency with this previous literature we have used the ‘energy chain’ terminology throughout.

Line 48: ‘present in most free living Bacteroidota’ – this is an overstatement. The citation #16 is a review where this system is briefly described along with all other bacterial motility systems. Is there a study of most free living Bacteroidota and their motility?

Response: The statement is made on the basis of gene occurrence in sequenced genomes. The text has been altered to indicate this. The best formal analysis of this is McBride and Zhu (*J Bacteriol* (2013) **195**: 270–278), and this is now cited at this point. A more comprehensive genome analysis is present in the thesis of Andreas Kjaer and this is also now cited at this point.

Line 49.

Line 57: citation # 24, 25 are used to support a statement on SprF – do these two articles have any experiments on SprF?

Line 55: “SprF is, in turn, thought to bind to a moving track”. To the best of my understanding none of the cited references specifically indicate that sprF could be the entity that links sprB to the track. However, given the fact that 1. SprB protects sprF from proteolytic cleavage in intact cells 2. co-IP experiments have shown direct interactions and 3. that SprB moves over a helical ‘track’ in combination suggests the possibility of SprB-sprF-track arrangement. But there is no direct fluorescently tagged SprF known to move helically neither is there any cryo-ET or cryo-EM data in support.

Since this sets the premise of the work, authors could write this clearly and cite appropriate and specific references for wider readership and be more transparent in their writing.

Response: Line 57 is the sentence quoted as line 55 so we take these comments together.

As noted by the Reviewer, these references do not have experiments that support the linkage of SprF to the track. However, the full sentence reads: ‘SprF is, in turn, thought to bind to a moving track structure at the inner face of the OM resulting in the propulsion of the attached adhesin along the cell body’. ‘Thought’ indicates that there is no direct evidence for this and that it is a model. This sentence is a reasonable, concise summary in the Introduction of the current model with citations. We have now added a citation that explicitly shows a model in which the adhesin carrier protein complex directly interacts with the moving track (Fig. 5d and Discussion text in Lauber et al., 2024). **Line .**

The Reviewer accepts that the SprB adhesin is bound to SprF, and that SprB moves helically, but then objects that ‘there is no direct fluorescently tagged SprF known to

move helically'. However, since SprF is in complex with SprB which moves helically, SprF MUST be moving helically. More to the point, even if we were to experimentally confirm that SprF moves helically this would not show that SprF is directly bound to the track. So the Reviewer's point here is unclear. Parenthetically we note that it is not possible to make functional fluorescent fusions to SprF – this is not surprising given that the exposed faces on either side of the membrane to which fusions could be generated are involved in interacting with partner proteins (SprB on the exterior, the gliding track on the interior).

The inference that SprF binds to the track is the most obvious and parsimonious hypothesis given the observations (all stated in the Introduction) that (i) the adhesin is anchored on the outside of the cell by the transmembrane protein SprF, (ii) that the adhesin is moved around the cell on the gliding track, and (iii) that the gliding track proteins are positioned on the inner face of the outer membrane. Indeed, what is the alternative hypothesis? If there were another component bound to SprF that in turn binds to the track, then it is very surprising that Gorasia and co-workers only identified SprF when pulling on SprB.

We accept the Reviewer's point that the premise of our experiments is that SprF directly interacts with the gliding track and that this has not been explicitly experimentally demonstrated before this work. We have, therefore, modified the paragraph at the end of the Introduction that introduces the experimental question to make it clear that the SprF-track interaction is only a proposal at this point. We also reworded the text to allow that we are investigating more generically the 'connection/linkage' between the two rather than explicitly a contact interface. Lines 79, 84, 94.

Line 73-74 – “contacts the outer ring” - none of the cited references show this via direct experimentation – at this point, it is mostly a very plausible hypothesis.

Response: Again, the text is not claiming that this is anything other than a hypothesis: 'In the gliding bacterium *F. johnsoniae* the PorKN-equivalent Hub proteins GldK and GldN35 are ASSUMED to form a similar ring structure, the outer rim of which contacts the gliding track.'

Line 117 – “removing the cysteine pair” – authors often use the word removing when they supposedly mean substituting. A consistent use of the correct indication is encouraged to remove ambiguity.

Response: These instances have been corrected.

Line 118 – “Thus, the twin-cysteine motif is required for SprF to make normal interactions with the gliding track” – the finding that substitution of cysteine residues leads to loss of sprB dynamics while retaining the export does not necessarily mean that the interaction of sprF with the “track” was lost.

Response: We cannot think of an alternative reasonable interpretation given that SprB is anchored to the cell through SprF, and the Reviewer does not suggest one.

Even when using this claim further it is imperative that the authors experimentally demonstrate that sprF associates with track proteins. Without establishing this as an

experimentally validated fact, the loss of function leading to abrogated adhesin tracking, can not be attributed to the cysteine subs.

Response: See responses above and below to the same objection.

Line 128– “Thus, SprF is still able to interact with the gliding tracks...” this partially contradicts the categorically stated inference in line 118. However, the larger issue still is the assumption that sprF interacts with track in the absence of direct experimental evidence.

Response: There is no contradiction. The previous sentence says that it is required to make ‘normal’ interactions with the track. The normal interaction is sustained, continuous helical movement. That is not what is observed here, so whatever is happening is no longer ‘normal’.

On the ‘larger issue’ see the response to the same point above.

Line 171 –” Thus, the Hub component GldK is the partner protein to which SprF is disulfide linked. Because SprF is in continuous movement along the gliding track, the covalent link between SprF and GldK implies that GldK is a component of the track rather than being located in a separate Hub complex as envisaged in the current gliding model”

There are several caveats to this conclusion:: 1. A successful probe of two different proteins with their cognate antibodies at the same band size after disulfide-crosslinking can not be an adequate proof that a direct interaction between these two proteins exists. Authors must confirm this interaction with more definitive methods such as mass-spec of the band-cutouts and peptide identification from them.

Response: We validate the disulfide bond between SprF and GldK by removing the cysteine residue in partner 1 and seeing the disulfide adduct detected for partner 2 disappearing. This is done for the GldK-detected adduct at this point in the text, and for the complementary SprF-detected adduct in the following paragraph. The Reviewer has completely overlooked this key point.

The data is quite clear already. However, to definitively substantiate this point we now include experiments (new Fig. 2c) in which we pull on SprF under denaturing conditions and demonstrate that it is covalently linked by a disulfide linkage to GldK.

2. It may be assumed by some educated and confident extrapolation (as indicated earlier in the comments) that sprB-sprF-track -connection might exist, but the claim that sprF moves along the track must be experimentally demonstrated.

Response: As noted above in response to an earlier comment since SprB forms a complex with SprF, and SprB is moving along the tracks (the Reviewer accepts both points), then SprF must also be moving along the tracks.

In addition, it would perhaps be too presumptuous to claim based on the assumption that GldK is a part of “track”. That claim, in its own regard must be substantiated by experimental evidence of the highest rigor, such as labelling GldK and showing it in situ to localize along the “track” via cryo-ET.

Response: Again it is quite clear that we are presenting this as a hypothesis. This hypothesis is developed more fully later in the paper and the multiple lines of evidence supporting it are laid out in the Discussion. It is not apparent what the alternative plausible hypothesis to explain the data would be – and the Reviewer does not put one forward. Crucially, however, we are not just presenting this model in isolation. At this part of the text we pointing out that a disulfide bond between SprF and GldK is **incompatible with the current model for the role of GldK**. This is key – we are falsifying the current model.

Line 179 – “substituted providing presumptive evidence that it is Cys343 that forms the disulfide linkage with SprF...” the current experimental evidence can only support the inference that the cys343 is essential to the function of GldK and cannot support the above claim– again an analytical method such as mass spec can be useful for more definitive answer.

Response: We know that the adducts are formed via a disulfide bond (they are reversed by DTT) so they must involve cysteine residues. We individually substitute all the cysteine residues in GldK (this is the next sentence after the one highlighted by the Reviewer) and only one abolishes adduct formation. So what other explanation could there possibly be for these observations than that the identified cysteine residue in GldK is involved in the disulfide bond to the partner protein? The other cysteines in GldK cannot be doing this.

If the reviewer is making the point that Line 179 is not by itself evidence for this disulfide link we agree – that is why the sentence contains the word ‘presumptive’ – and why we prove it in the next sentence, as just outlined.

Line 182 – “...albeit with lower efficiency than in the wild-type strain (Fig. 3d)” the ‘lower efficiency’ claim cannot be accurately made especially given that the intensity of both of the ProtK and TX100 bands in C343S as compared to the WT is higher. In addition, there is greater variation seen in the GroEL loading control bands”. This is of course just the visual interpretation and authors can perhaps supplement quantification with densitometric analysis.

Response: The Reviewer has misunderstood what is shown in this panel. The conclusion does not depend on band intensity between lanes but the relative band intensities within a lane. In the C343S mutant there are both protease protected and unprotected bands – this means that some of the adhesin has been exported (unprotected) and some has not (protected). In the wild type strain there is only exported (unprotected) adhesin.

Line 192 – “...Cys343 on their mobility are equivalent to those of removing the SprF twin-cysteine motif. This is as expected if these two groups of cysteine residues are partners in the same disulfide bond”

Response: There is no point made here by the Reviewer, so no response is needed.

Line 204 – “Our interpretation of this observation is that Cys324 forms an intramolecular disulfide bond with Cys326, and that this bond is then attacked by a cysteine thiol on GldK

to form the SprF-GldK disulfide link” :: this sounds like a very plausible explanation. If this can not be backed by experimental evidence such as from advanced analytical methods, it may at least be helpful to cite example(s) of other proteins with twin cys motifs, where such intermolecular disulfide bonds may have been demonstrated.

Response: This is standard disulfide chemistry that can be seen to occur during intramolecular disulfide bond formation and isomerisation catalysed by systems in the bacterial periplasm (e.g. Dsb systems in *E. coli*). However, in these cases the formation of intermolecular disulfides that occurs between the target protein and the biosynthetic proteins are transient and catalytic. As we note in the Discussion, there are almost no other examples of a permanent intermolecular bond formed between two periplasmic proteins. We now note another example. However this does not appear to be produced by the attack of a lone Cys in one protein on the disulfide bond in the partner protein as we propose for the GldK-SprF/PorG linkage.

Line 587.

Line 290– what kind of analysis? It would be a good context if the logic are explained here.

Response: Sequence analysis. This has now been clarified in the text.

Line 331 – “This adduct was absent in a gldK deletion mutant (Fig. 4f) confirming that GldK is the partner protein to which PorGFj is disulfidelinked.” – again, same comment as before- cooccurrence of bands in presence/absence of NEM on gels is inadequate experimental evidence for this claim. A strong analytical method must confirm this finding.

Response: We go on in the next sentence to show that the band disappears in the appropriate GldK Cys substitution but not on removal of the remaining Cys substitution, as for the case of SprF. As for SprF, we then carry out the reciprocal analysis with mutations in PorG or its Cys residues and blotting with GldK. As with the linkage to SprF the Reviewer has missed the main evidence for this deduction.

As with the SprF linkage to GldK the data is quite clear already. However, and again, to definitively substantiate this point we now include experiments (new Fig. 4g) in which we pull on PorG under denaturing conditions and demonstrate that it is covalently linked by a disulfide linkage to GldK.

Line 383– the sentence does not sound complete or has an extra word (proteins in the OM protein)?

Response: The typo has been corrected.

Line 386 – to reiterate, GldK being part of the track in Flavobacterium would necessitate strong experimental evidence including but not limited to its distribution across the cell in a helical fashion as the “track”.

Response: We in turn reiterate the discussion of the evidence in the Discussion and our responses above. But to set this out:

- SprF is being moved by the tracks (we have shown above that the Reviewer's objections to this carry no weight).
- SprF is (apparently permanently) disulfide linked to GldK (we have shown above that the Reviewer's objections to this carry no weight).
- Thus GldK is being moved by the tracks.
- Thus GldK is NOT part of a Hub complex as in *P. gingivalis* (the current hypothesis), because this would restrict GldK to a very small area of the cell.
- Given that GldK is moving with the tracks, has the same cellular location as the tracks, and is in complex with GldN which is in contact with the motors that drive movement of the tracks (various Cascales group papers), the logical deduction is that GldK is part of the tracks.
- This deduction is supported by high confidence AlphaFold modelling of the track as a complex between GldK, GldN, and the proposed track protein GldJ, and this structural model is highly consistent with the experimental structure of the homologous Hub complex we obtain here.

Once again it is necessary to ask what the alternative plausible hypothesis is that is consistent with the crosslink between SprF and GldK – the Reviewer does not provide one.

The other Reviewers apparently have no problems with any of our logic.

Line 386– “This implies that...” this is an interesting thought, especially having demonstrated and discussed (albiet with inadequate evidence) the parallel between ‘pulling’ cargo off the translocon and transporting an adhesin on the ‘track’, however this claim requires a lot more and stronger experimental evidence as indicated prior in the comments.

Response: See our responses to this point multiple times above.

Line 425 – “The contacts between...” it will be helpful if this sentence was split to improve understanding.

Response: Altered as suggested.

Line 437.

Line 470 –“dynamic” can be interpreted in multiple ways here. Do authors mean dynamically changing interaction in the cell between the two proteins or do they mean different ways the interaction may occur in different bacteria? Some clarification could be more useful to understand the authors’ point of view.

Response: The former. The word `dynamic’ would not be used to describe the Reviewer’s second suggestion.

Line 476 – “Mini-track” – As suggested in the comments before, the bonafide status of “track” or “mini-track” should be reserved until adequately backed by experimental evidence. This mini is thus far called as a ring – Porphyromonas gingivalis does not have gliding motility – why does it need a mini track?

Response: `Mini-track' is shorthand that elegantly encapsulates the proposed evolutionary and structural relationship between gliding tracks and the Hub complex, and at this point in the text summarizes the rationale for wanting to determine the structure of the *P. gingivalis* Hub complex.

The reason that *P. gingivalis* retains a track is to continue to power the T9SS (this is set out in the Discussion). More specifically, the proposal would be because it has to move PorG and carrier proteins (we have now specified this in the Discussion, Line), just as the gliding track does. And we know that PorG is essential for T9SS and, from this work, that it is disulfide bonded to the Hub/mini-track.

General comments:

The AF3-prediction that GldK is part of the 'track' - AF3 models of multiprotein complexes can provide a hypothesis and they need to be experimentally verified – I suggest the authors present their model as an alternative hypothesis and test it in future via rigorous experimentation.

Response: It is already presented as a hypothesis. Nowhere in this section is this presented as fact. Even the title of this section is `MODEL for the *F. johnsoniae* gliding track structure'.

Authors could mention the purpose of Triton X-100 (intact cells vs cell lysate) in either methods section or in one of the figure legends. This will improve on the accessibility of the work for all reader-levels. Particularly, not using it in the experiment of fig.1 but everywhere else - would get a better context. The cited references does not explain this either so this would be a great addition.

Response: We have added this information to the appropriate Figure legends.

The figure legends should show the number of trajectories or sprB foci analyzed for TIRF experiments, and associated statistics. It would also be ok- perhaps better- to have a histogram of relevant property as additional figure panels, for eg., but not limited to, reversal frequency in different strains in fig4g (C225S)

Response: The data are exemplar. All trajectories detected are shown except where edited out as noted in the Figure legends. We just note that mid-cell reversals are often seen in the *porG* mutants which is not behaviour that is normally seen in the parental strain. We do not attempt to quantify this.

Editorial:

The authors might want to double check the doi URL's. Some of them did not work for me.

Response: The journal style does not have these URLs. We have updated the referencing format for the journal style.

Reference 20 – what is this? There is nothing but the name of authors, title, year and I was not able to find this article online.

Response: This is not yet available in print or preprint so we have deleted this reference.

Reviewer #2 (Remarks to the Author):

This manuscript by Liu and colleagues explores the structure and function of the type nine secretion system (T9SS) and of the related Bacteroidota gliding motility machinery. Using genetic, biochemical, and structural studies the authors made the unexpected discovery that key outer membrane gliding (SprF) and T9SS (PorG) proteins form disulfide bonds to cysteines on GldK. In P. gingivalis, PorK (the homolog of GldK) is part of a ring complex that interacts with the PorLM (GldLM) motor to energize secretion. F. johnsoniae GldK is involved in both gliding and in T9SS-mediated secretion. The authors demonstrate that formation of the disulfide bonds are critical for proper function of the gliding and T9SS machineries. In the case of the SprF-GldK disulfide bonds, these strongly attach the outer membrane gliding motility protein SprF to GldK on the helical motility tracks that underlie the outer membrane. It was previously shown that the external motility adhesin, SprB, that is propelled along helical paths along the cell surface, interacts strongly with the outer membrane protein SprF. The results presented here provide the link between SprB and the internal track system that in turn interacts with the energizing GldLM motor in the cytoplasmic membrane to result in gliding. The story is similar for the T9SS protein PorG. The PorG disulfide bonds with GldK/PorK, and the PorG cysteines needed for this are important for proper T9SS-mediated secretion in both P. gingivalis and in F. johnsoniae. The authors also used cryoEM to explore the structure of the P. gingivalis PorKN ring complexes. P. gingivalis is nonmotile, lacks GldJ, and appears to have a simpler GldK/PorK complex architecture than does the motile F. johnsoniae, where direct observation of the complex at high resolution has been difficult. The authors used the P. gingivalis PorN/PorK structures that they observed to model the more complex tracks that appear to be formed by F. johnsoniae GldJ, GldK, and GldN. The authors present an intriguing model where the PorKN rings, which are only involved in P. gingivalis T9SS-mediated secretion, are a simplified version of the helical track structures formed in F. johnsoniae, where the proteins appear to function directly in both gliding and in secretion.

The results are convincing and are important in understanding the structure and functioning of the T9SS and the gliding motility machinery. Since both T9SS-mediated secretion and motility have been linked to virulence, and also impact the functioning of many bacteria of environmental or biotechnological relevance, they may also lead to practical applications in the future. The manuscript is well written and presents interesting, surprising, and important findings. The results greatly increase our understanding of gliding motility, by providing evidence for a direct chain of proteins linking the substratum over which the cell glides, with the gliding motor in the cytoplasmic membrane. They also provide a model for how some of the core components of the T9SS

participate in both cell movement and in protein secretion. I have only minor suggestions for improvement, as listed below:

1) Is it known how SprF interacts with SprB? From previous studies this probably involves the C-terminal region of SprB, but I don't know if the exact regions of interaction are known. I am not suggesting additional experiments on this point for this paper. Just request to add information if it is already known/published.

Response: As the Reviewer notes, and as cited in the manuscript, previous studies have shown that the most C-terminal portion of the molecule (last 368 amino acids which includes the Type B C-terminal Domain (CTD) of SprB) is sufficient to attach SprB to SprF (*Int J Mol Sci* (2022) **23**: 5681). The reasonable assumption from this observation has been that within this region the interaction is between the CTD and the carrier protein as has been structurally established for Type A CTD-carrier protein interactions (*Nat Microbiol* (2024) **9**: 1089-1102) and this is supported by high confidence AlphaFold modelling (our unpublished observations). Since the information is not known for certain, and because it does not bear on the results in the manuscript, we have steered clear of commenting on this or showing this explicitly in the schematic figures.

2) p 2 line 21 "moving internal track structure which propels them through the membrane." and p 26 line 610 "These track movements would then drag the disulfide-linked partner proteins through the OM."

"Through the membrane" may suggest to some that they are moving 'across' the membrane (from inside to out for example). Would 'along the membrane' be better? I am not sure. Choose what you think works best here.

Response: We have altered these to read 'laterally through' rather than 'through'.
Lines 22 and 631.

In addition, for p 2 line 21, 'which' should be 'that'.

Response: This has been corrected.

3) p 3 line 28

referring to Bacteroidota the authors state "They are the most abundant Gram-negative commensals in the human gut and "

Are they always the most abundant G- commensals? If yes, then no change needed. If this is not certain, then I suggest change to something like:

"They are often the most abundant Gram-negative commensals in the human gut and "

Response: The Reviewer is correct, since this is not always true under certain disease conditions. We have amended with the qualification 'normally'.
Line 29.

4) p 5, 6 line 96 - 97

"Substituting either or both of the (SprF) cysteine residues abolished gliding (Fig. 1c)."

Fig 1c shows colonies that either spread because of gliding motility (wild type), or that did not spread or at least spread much less (sprF mutants). It is clear that motility was affected by the mutations. What is not entirely clear is that motility was completely abolished. The authors did not examine movement of individual cells over surfaces but rather looked at the collective spreading. It is known that disruption of F. johnsoniae sprF or of other 'spr' genes typically results in nonspreading colonies (hence the name 'spr'), but the individual cells retain some limited ability to glide on some surfaces. This is in contrast to disruption of gld genes, which typically results in complete inability of cells to move/glide. The sprF mutants constructed by the authors are certainly defective in gliding, but i suspect that like other spr mutants, they retain the ability to make slight movements on some surfaces. With this in mind, 'abolished gliding' seems unnecessarily strong.

similarly, in other places in the text:

p 11 line 230

"Both cysteine residues in the C-tail of SprF are essential for gliding (Fig. 1c)"

'essential for gliding' seems too strong. Perhaps 'important for gliding' would be more accurate.

p 39, (Fig 1 title) "The cysteine residues in the periplasmic C-tail of SprF are essential for SprB

movement and gliding motility"

This figure title is not accurate for the reasons described above. The figure actually demonstrates movement of some SprB molecules along the cell surface of some of the mutants that lack the sprF cysteine residues. Clearly, the movement of SprB is affected by the loss of these SprF cysteines, but 'essential for SprB movement' seems too strong, and actually incorrect.

As indicated above, there are many ways to fix these phrases, such as by indicating 'important for gliding' etc. rather than 'essential'!

Also, note that complete loss of gliding is not expected for sprF mutants. SprF is only known to interact with one adhesin, SprB. Deletion of sprB does not completely eliminate gliding. There are other adhesins that can replace SprB and function in gliding on some surfaces. Presumably they interact with the track in a similar way. Note that these suggestions do not question the importance of your results, just the accuracy of your language here.

Response: The Reviewer is correct that the gliding phenotypes have only been assessed for gliding on agar surfaces and not on glass. As the Reviewer notes, the mutants might be expected to make `slight movements' on some surfaces due to non-SprB adhesins, and so technically only SprB/F-dependent motility has been abolished (though whether

cells that do not progress from their starting position should be classed as 'gliding' is an interesting question). As suggested, we now take these subtleties into account by using 'important for gliding' rather than 'essential for gliding' or by specifying that the gliding is on agar.

The Reviewer is correct in noting that substituting single Cys residues in SprF does not necessarily block all adhesin movements and so we have corrected the Fig. 1 title.

5) p 9 line 171

"Because SprF is in continuous movement along the gliding track, "

It might be better to indicate 'Because SprF appears to be in continuous movement along the gliding track,' .

You only monitored the movement of SprB. Your results and previous results do indicate a strong interaction of SprB and SprF, so i think you are correct, but perhaps the language should be more careful here, since you did not directly observe 'continuous movement' of SprF.

Response: We have changed the text as suggested by the Reviewer.

6) p 14 line 301 *"an observation which is consistent with the hypothesis that PorG is a disulfide-linked"*

'which' should be changed to 'that'.

similarly, on p 31 line 706, "0.5ml fractions which were then either supplemented", 'which' should be changed to 'that' (or you could add comma after 'fractions' (fractions, which...), but i think 'that' makes more sense here).

Response: These have been corrected.

7) p 52, lines 1064-1065. *Extended data fig 3.*

"and a T9SS-targeting C-terminal domain (CTD). whole cell and supernatant samples analyzed by anti-mCherry immunoblotting."

Needs attention. Period in wrong place? Should this all be one sentence?

Response: This has been corrected.

8) *Ref 20. Journal name?*

Response: This is not yet available in print or preprint so we have deleted this reference.

9) Ref 37. Thesis? if so, indicate.

Response: Yes. This has been corrected.

Reviewer #3 (Remarks to the Author):

Response: No response required.

Reviewer #4 (Remarks to the Author):

In this manuscript, Liu and co-workers report a biochemical and structural characterisation of the T9SS-mediated gliding mechanism in Bacteroidota. This work significantly transforms our understanding of both gliding motility, and T9SS-mediated protein transport. Specifically, it very elegantly demonstrates that the protein GldK is a component of the track, employed by both gliding and secretion machineries; and that it binds to the adhesion adapter protein SprF through disulphide bonds. This mechanism supersedes previous models, and is supported by extensive validation through structure-function analysis. Accordingly, I am highly supportive of its publication in Nature Communications (though I do wonder if it would be best suited for Nature Microbiology, considering its broad impact for our understanding of bacterial motility and secretion), and I only have a few minor comments for suggested edits.

- I think the term "new" or "novel" throughout the manuscript (in the title, the legend for Fig 6, and elsewhere) is perhaps not appropriate, and will likely be challenged by the journal guidelines anyway. So I would recommend the authors to remove these terms, and refer to "updated" and/or "refined" models as appropriate.

Response: We have revised the manuscript paying particular attention to this point and removed the words `novel' and `new' at all points, including the title and abstract. In one case we have substituted the term `unprecedented' which we use in the dictionary sense of the word to succinctly capture the nature of the entity rather than as a claim of priority i.e. this is a factual statement rather than a judgement.

- I recommend that the authors include an additional supplementary video, illustrating the structure of the PorKN complex, and the various interfaces observed between these proteins. An additional video showing their fit within the low-resolution cryoET map, could also be helpful.

Response: This has been done. These videos are now included as Supplementary Videos 5 and 6.

- The results section makes the structural analysis of the PorKN complex sound absolutely trivial, but the methods section (particularly Supplementary Figure 6 - labeled as figure 7 in the methods section!) show that it was actually rather complex. I recommend including a few additional sentences in the corresponding part of the manuscript, describing the cryo-EM data processing process. Considering the size of the complex, I would have expected major preferred-orientation issue; was this the case? If so, how was it dealt with? Similarly, the authors refer to various stoichiometries for the complex, from 31-fold to 35-fold. How were these identified? Could these be separated through classification? Additional data could be helpful here.

Response: The preferred orientation problem meant that reconstruction in C1 yielded poor volumes with a preponderance of views being top-down. Since the symmetry could be assigned for the majority of the top-views by counting, the 33-fold symmetric top views (the vast majority) were selected as were all tilts and side views and these were used for further processing. Further 3D classification in D33 symmetry was used to select out those particles that were genuinely of this symmetry (presumably the bulk of particles removed came from the tilts and side views, since no initial visual selection had been applied to these). Following symmetry expansion and focusing in on the trimer-masked region, the view distribution was not problematic, as can be assessed from the cFAR score seen in the workflow. We have added to the methods to expand on this. For particles with other symmetries, the low number of particles at each symmetry precluded high resolution reconstruction of these objects.

The figure number typo has been corrected.